# Antigen presentation by lung epithelial cells directs CD4+ T_{RM} cell function and regulates barrier immunity

Anukul T. Shenoy [1], Carolina Lyon De Ana [1,2], Emad I. Arafa[1,3], Isabelle Salwig[4], Kimberly A. Barker[1,2], Filiz T. Korkmaz[1], Aditya Ramanujan [1], Neelou S. Etesami [1,2], Alicia M. Soucy [1], Ian M. C. Martin[1], Brian R. Tilton[5], Anne Hinds[1,3], Wesley N. Goltry[1], Hasmeena Kathuria[1,2], Thomas Braun [4], Matthew R. Jones[1,3], Lee J. Quinton [1,2,3,6], Anna C. Belkina [1,5,6] & Joseph P. Mizgerd [1,2,3,7✉]

Barrier tissues are populated by functionally plastic CD4+ resident memory T (T_{RM}) cells. Whether the barrier epithelium regulates CD4+ T_{RM} cell locations, plasticity and activities remains unclear. Here we report that lung epithelial cells, including distinct surfactant protein C (SPC)$^{low}$MHC$^{high}$ epithelial cells, function as anatomically-segregated and temporally-dynamic antigen presenting cells. In vivo ablation of lung epithelial MHC-II results in altered localization of CD4+ T_{RM} cells. Recurrent encounters with cognate antigen in the absence of epithelial MHC-II leads CD4+ T_{RM} cells to co-express several classically antagonistic lineage-defining transcription factors, changes their cytokine profiles, and results in dysregulated barrier immunity. In addition, lung epithelial MHC-II is needed for surface expression of PD-L1, which engages its ligand PD-1 to constrain lung CD4+ T_{RM} cell phenotypes. Thus, we establish epithelial antigen presentation as a critical regulator of CD4+ T_{RM} cell function and identify epithelial-CD4+ T_{RM} cell immune interactions as core elements of barrier immunity.

[1] Pulmonary Center, Boston University School of Medicine, Boston, MA 02118, USA. [2] Department of Microbiology, Boston University School of Medicine, Boston, MA 02118, USA. [3] Department of Medicine, Boston University School of Medicine, Boston, MA 02118, USA. [4] Department of Cardiac Development and Remodeling, Max-Planck-Institute for Heart and Lung Research, Member of the German Center for Lung Research (DZL), Bad Nauheim, Germany. [5] Flow Cytometry Core Facility, Boston University School of Medicine, Boston, MA 02118, USA. [6] Department of Pathology and Laboratory Medicine, Boston University School of Medicine, Boston, MA 02118, USA. [7] Department of Biochemistry, Boston University School of Medicine, Boston, MA 02118, USA. ✉email: jmizgerd@bu.edu

Tissue-resident memory T ($T_{RM}$) cells operate as front-line adaptive immune sentinels for barrier tissues, such as the skin, lungs, vagina, and intestine[1,2]. Barrier tissues experience unavoidable, transient, repeated exposures to environmental antigens. $T_{RM}$ cell formation and responses depend on diverse natures of antigenic stimuli, types of antigen-presenting cells (APC), and costimulatory signals and cytokines within the challenged tissue[3,4]. These signals diversify $T_{RM}$ cell responses beyond their initial phenotype specification, yielding T cell plasticity that enhances anti-microbial immunity and tissue barrier integrity[3,5,6]. Because $T_{RM}$ cells and T cell plasticity have been linked to barrier tissue pathologies like asthma, psoriasis, and inflammatory bowel disease (IBD)[7–10], tissue-resident regulators of $T_{RM}$ cell plasticity have become an important knowledge gap.

As direct interfaces between the environment and the body, epithelial cells are poised to be critical regulators of barrier immunity[4,11]. Barrier epithelial cells recruit and maintain $CD8^+$ $T_{RM}$ cells near the sites of antigen encounter, and reactivate them in the tissues via local antigen presentation[12,13]. In contrast to $CD8^+$ $T_{RM}$ cell effects, epithelial roles governing $CD4^+$ $T_{RM}$ locations, plasticity, and activities remain speculative. Distinct barrier epithelial cells, such as alveolar type 2 (AT2) cells of the lungs[14], $Lgr5^+$ stem cells of the small intestinal crypts[15], enterocytes of the villi[16], and keratinocytes of the skin all express MHC-II[17]. Although the only known function of MHC-II is antigen presentation to $CD4^+$ T cells, whether barrier epithelial cells direct $CD4^+$ $T_{RM}$ cell activities during exposures to relevant cognate antigens remains speculative.

*Streptococcus pneumoniae* (*Spn*) is a leading cause of pneumonia and infectious deaths worldwide[18]. Serotype replacement and antibiotic resistance have contributed to the classification of *Spn* as a priority pathogen by WHO in need of better antibiotics and vaccines. The protection offered by *Spn*-specific $CD4^+$ $T_{RM}$ cells is $T_H17$ polarized and serotype-independent[19], representing a potential avenue for vaccination against all *Spn* serotypes. Especially considering that up to 65% of children and ~10% of adults are exposed to this pathobiont at any given time[18], understanding the biology of *Spn*-specific $CD4^+$ $T_{RM}$ cells is imperative to improving the outcomes of recurrent *Spn* exposures.

In this work, we postulate that lung epithelial cells (LECs) mediate antigen presentation to pulmonary $CD4^+$ $T_{RM}$ cells in order to regulate their localization and activities after *Spn* infection. We define molecular factors involved in this communication axis and show a physiological role of LEC MHC-II related to $CD4^+$ $T_{RM}$ cell plasticity and immunity.

## Results

**Identification of SPC^low^MHC^high^ LECs.** We sought to identify LECs with antigen-presenting abilities during homeostasis. Because MHC-II has been localized to AT2 cells[14] and AT2 cells have no known cell-specific surface markers useful for flow cytometry, we used mice expressing GFP under control of human surfactant protein C (SPC) promoter (SPC-GFP mice)[20] in order to discriminate the SPC-GFP^high^ AT2 cells. At baseline, LECs from elastase digested lungs of (Fig. 1a) separated into two clusters based on MHC-II profiles. The MHC-II^low^ cluster was largely comprised of airway LECs (i.e., club and multiciliated cells) with minimal contribution of alveolar type 1 (AT1) cells (Fig. 1a and Supplementary Fig. 1a). The MHC-II^high^ cluster was comprised of SPC-GFP^high^ AT2 cells and a comparable-sized fraction of SPC-GFP^low^ cells (Fig. 1a and Supplementary Fig. 1a). Interestingly, the SPC-GFP^low^ cells expressed more MHC-II than AT2 cells, almost comparable to levels observed on $CD45^+CD24^+MHC-II^+$ professional APCs (Fig. 1b and Supplementary Fig. 1b). These SPC-GFP^low^ cells exhibited smaller forward scatter, comparable to club

cells (Fig. 1c), and higher β4 integrin (Fig. 1d) and stem cell antigen Sca-1 (Fig. 1e) when compared to AT2 cells. We sought to define the SPC-GFP^low^ cells using mRNA for lineage-specifying markers. The SPC-GFP^low^ cells coexpressed markers characteristic of AT2 cells (*Spc*) and club cells (*Scgb1a1*), but not markers for multiciliated cells (*Foxj1*) or AT1 cells (*Aqp5*) (Fig. 1f). To further characterize this LEC type, we used bronchioalveolar stem cell (BASC) reporter mice that express YFP and mCherry inserted into *Spc* and *Scgb1a1* loci, respectively[21] (Fig. 1g). These mice confirmed a population of SPC^low^ cells with high levels of MHC-II, and revealed that they were distinguishable from club cells (SCGB1A1^high^) that were MHC-II^low^, bronchioalveolar stem cells[22] (BASC; SCGB1A1^medium^ SPC^medium^) that were MHC-II^medium^, and AT2 cells (SPC^high^) that were MHC-II^high^ (Fig. 1g). The SPC-YFP^+^ cells were consistently found within alveoli and not in conducting airways, suggesting an alveolar location for SPC^low^ LECs (Fig. 1h). Transmission electron microscopy (TEM) of sorted SPC^low^ LECs revealed these cells contain both the AT2 cell-defining lamellar bodies, as well as electron-dense secretory vesicles characteristic of club cells (Fig. 1i). Lamellar bodies were exclusively observed in alveoli and never in airways upon inspection of >100 anatomically distinct areas of C57BL/6J mouse lungs (Supplementary Fig. 1c), leading us to conclude that SPC^low^MHC-II^high^ cells localize to alveoli rather than conducting airways. The SPC^low^ cells also possessed a transcript profile, including Sox2, MHC-I molecule *H2-K1*, long non-coding RNA *Aw112010*, and cell cycle regulator *Cdkn1a* (Supplementary Fig. 1d), reminiscent of distal lung stem cells[23]. Thus, these data suggest a discrete set of SPC^low^ LECs that express high amounts of MHC-I and MHC-II, exhibit transcriptional features of lung progenitor cells, and surface marker, transcriptional, and structural characteristics distinct from conventional LEC types, hereafter referred to as SPC^low^MHC^high^ LECs.

**LECs exhibit anatomically segregated and temporally dynamic APC activities.** To establish if antigen presentation is a feature of LECs, we examined their antigen uptake and processing abilities by intratracheal instillation of DQ-Ovalbumin (DQ-OVA). Consistent with antigen uptake and proteolytic degradation of DQ-OVA to release the quenched fluorophore, we observed fluorescence in all LECs at baseline and after infection with *Spn* serotype 19F (Fig. 2a and Supplementary Fig. 1a). Because antigen processing abilities were observed in cells with very low MHC-II at baseline, we tested whether MHC-II levels were altered in infection settings where antigen presentation becomes necessary. Indeed, Sp19F infection increased MHC-II on MHC-II^low^ LECs while further enhancing its expression on distal LECs, which remained the highest MHC-II expressers at all times (Fig. 2b). Further supporting APC functions, all LECs sorted from infected lungs displayed cell-specific patterns of MHC-II related accessory genes (Supplementary Fig. 2a) and activated antigen-specific $CD4^+$ T cells in MHC-II dependent fashions in ex vivo co-cultures (Supplementary Fig. 2b). Finally, to explicitly establish LECs as APCs, we generated mice lacking MHC-II in LECs by crossing H2-Ab^fl/fl^ mice[24] to Nkx2.1^Cre-ERT2^ mice[25] (henceforth MHC-II^ΔEpi^) (Fig. 2c). Administration of tamoxifen efficiently deleted all baseline or induced MHC-II on LECs (Supplementary Fig. 2c, d) while leaving MHC-II intact on other cell types (Supplementary Fig. 2e). Stimulation of antigen-specific $CD4^+$ T cells by antigen-fed LECs was compromised by MHC-II gene targeting (Fig. 2c). Thus, our results reveal that all LECs respond to infection by increasing MHC-II on their surface, and LEC MHC-II mediates antigen presentation to $CD4^+$ T cells.

Antigen presentation is modulated by costimulatory signals, so we examined if such signals were expressed by which LECs and when. Repeated exposures to *Spn* induces formation of protective

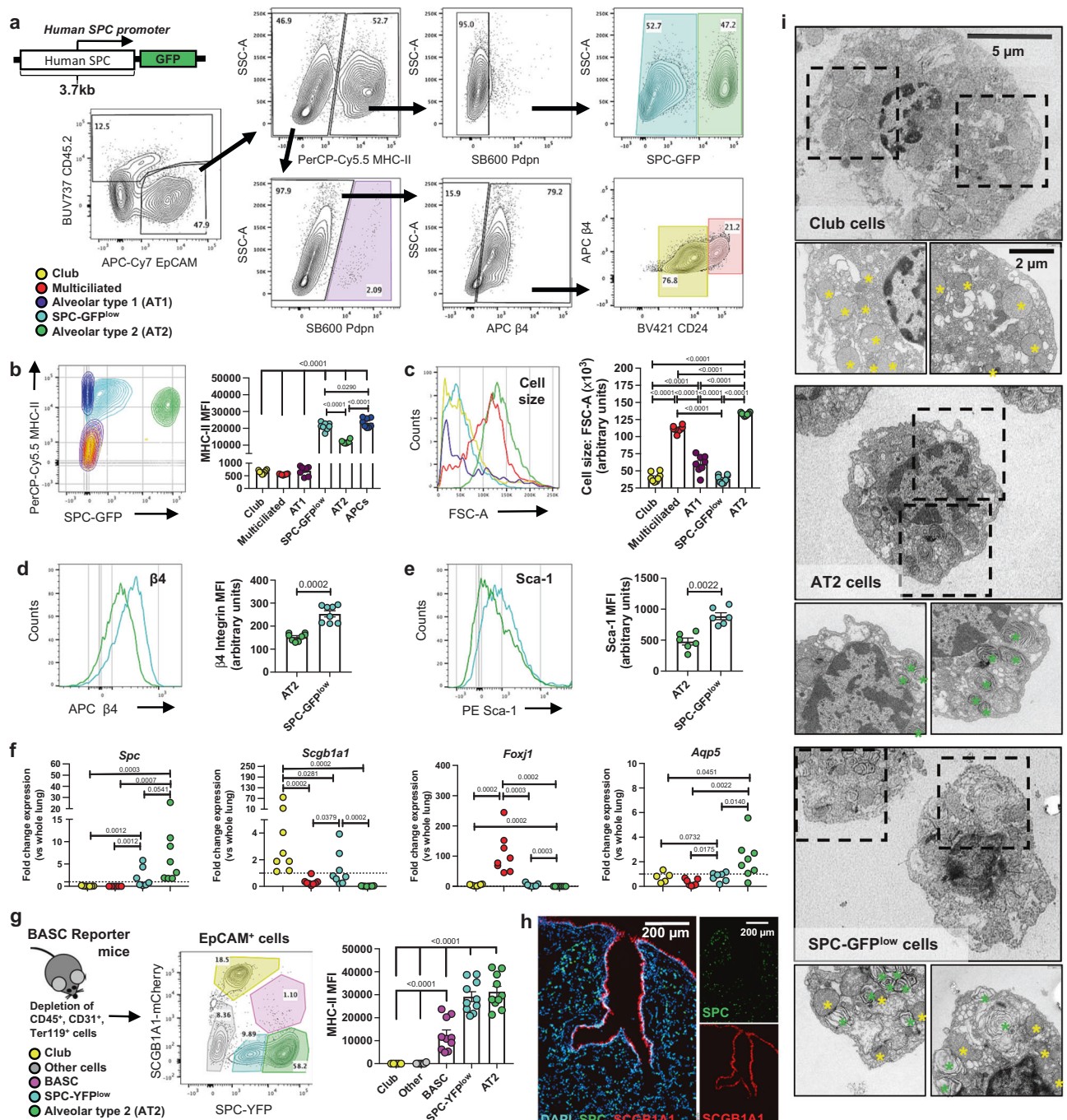

**Fig. 1 In addition to AT2 cells, MHC-II is highly expressed by SPC^lowMHC^high LECs. a** Genetic construct of SPC-GFP mice and gating strategy to identify major LECs including club (*yellow*), multiciliated (*red*), alveolar type 1 (AT1, *purple*), SPC^low (*blue*), alveolar type (AT2, *green*) lung epithelial cells in SPC-GFP mice. **b** Contour plot and quantification of LEC MHC-II. CD45^+CD24^+MHC-II^+ APCs depicted in *dark blue*. One-Way ANOVA with Holm-Sidak' multiple comparison test. **c** Histogram and quantification of LEC cell sizes, One-Way ANOVA with Holm-Sidak's multiple comparison test. Histogram and quantification for **d** airway epithelial marker β4 integrin and **e** stem cell antigen, Sca-1 by SPC-GFP^low and AT2 cells, two-tailed Mann–Whitney test. **f** mRNA levels of select lineage-defining transcripts in sorted LECs, two-tailed Mann–Whitney test. Sorting strategy is the same as gating strategy in Fig. 1a. **g** Contour plot for identification of distinct LECs in bronchioalveolar stem cell (BASC) reporter mice and quantification of LEC MHC-II, One-Way ANOVA with Tukey's multiple comparison test. Major LECs including club (*yellow*), BASC (*magenta*), SPC^low (*blue*), alveolar type (AT2, *green*), and other cell (gray) are indicated. **h** Representative immunofluorescent micrograph showing anatomical location of SPC-YFP^+ cells (*green*) and SCGB1A1-mCherry^+ (*red*) with DAPI (*blue*) used as counterstain. Data is representative of $n = 2$ mice, 2 experiments. **i** Representative transmission electron micrographs for LECs sorted and pooled from $n = 3$ SPC-GFP mice. Experiment was repeated twice. Yellow asterisks indicate electron-dense secretory vesicles and green asterisks depict lamellar bodies. All data have $n \geq 5$ mice, two independent experiments. All data are presented as mean ± SEM.

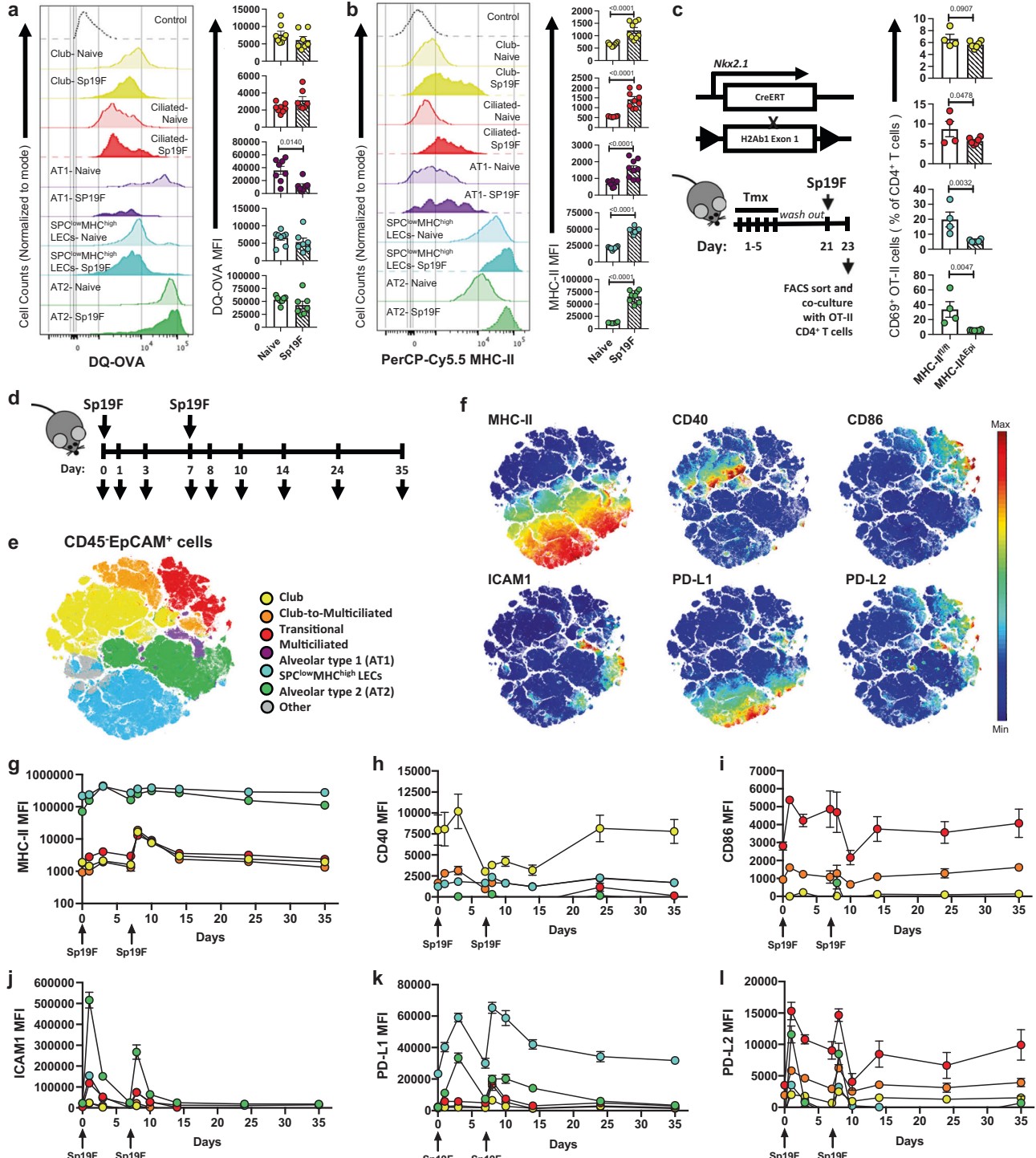

**Fig. 2 LECs display anatomically segregated and temporally-dynamic APC activity. a** Histogram and quantification of in vivo DQ-ovalbumin (DQ-OVA) uptake and processing abilities of LECs from naïve and Sp19F-infected mice 48hours postinfection (hpi), two-tailed Mann–Whitney test. **b** Histogram and quantification for LEC MHC-II from naïve and Sp19F-infected mice 48hpi, two-tailed Mann–Whitney test. **c** Genetic construct and experimental timeline of MHC-II$^{\Delta Epi}$ mice. Quantification of OT-II CD4$^+$ T cell activation by MHC-II$^{fl/fl}$ and MHC-II$^{\Delta Epi}$ LECs sorted from Sp19F-infected mice 48 h postinfection (hpi), one-tailed Unpaired $t$ test. Sorting strategy in Supplementary Figure. 2c. Note, different scales for Y-axes in Fig. 2a–c highlight that all LECs are capable of antigen presentation albeit in different amplitudes based on their anatomical location. **d** Schematic of experimental timeline. **e** opt-SNE plot depicting 6 LEC metaclusters identified from ~2 million LECs isolated from $n = 84$ mice across nine timepoints each with two independent experiments. **f** Heat map depiction for expression of APC-related molecules mapped onto opt-SNE projection. Temporal quantification for **g** MHC-II, **h** CD40, **i**. CD86, **j** ICAM1, **k** PD-L1, and **l** PD-L2 across distinct LECs. Two- Way ANOVA with Fisher's LSD test statistics for Fig. 2g–l are provided in Supplementary Table 1- 6. All data have $n \geq 4$ mice, two independent experiments. All data are presented as mean ± SEM.

lung CD4$^+$ T$_{RM}$ cells[19]. We collected LECs from multiple time-points after *Spn* infections (Fig. 2d) and subjected them to a 17-parameter high-dimensional multispectral flow cytometry (MSFC) panel that included antibodies to MHC-II, costimulatory molecules (CD80, CD86, CD40, ICOS-L), cell adhesion molecules with costimulatory functions (ICAM1 and VCAM1), and coinhibitory molecules (PD-L1 and PD-L2). Approximately 2 million CD45$^-$EpCAM$^+$ LECs from 84 C57BL/6J mice across timepoints were concatenated, projected into 2-dimensional optimized t-distributed stochastic neighbor embedding (opt-SNE) space[26] and subjected to unsupervised clustering using Phenograph algorithm[27]. This empowered us to (i) identify distinct LEC clusters based on expression profiles integrating all measurements rather than individual surface markers, and (ii) quantify the relative abundances of these clusters over time. Phenograph clustering identified 34 distinct LEC clusters that aggregated into 6 metaclusters corresponding to each LEC type (Fig. 2e and Supplementary Fig. 3a, b). One metacluster did not cleanly correspond to recognized LEC types and shared club and multiciliated features likely representing a club-to-multiciliated transitional state (clusters 14, 18, 19), consistent with our timeline encompassing lung epithelial damage and regeneration[28]. Opt-SNE visualization confirmed the identity of these metaclusters (Supplementary Fig. 3c)[26]. As expected, some clusters were products of dynamic signaling events and not all clusters were present simultaneously; instead within each metacluster certain clusters dominated at distinct time points (Supplementary Fig. 3d). These observations demonstrate that LECs undergo dynamic modulation of surface APC phenotypes during microbial infections concurrent with the formation of lung CD4$^+$ T$_{RM}$ niches.

Next, we tracked the surface expression levels of T cell-directed molecules on the LEC metaclusters (Fig. 2e) and discovered that expression of costimulatory/ coinhibitory molecules on LECs was dynamic and highly cell type-specific (Fig. 2f–l and Supplementary Table 1–6). Club cells and multiciliated cells consistently expressed higher levels of T cell costimulatory CD40 (Fig. 2f, h) and CD86 (Fig. 2f, i), respectively, compared to any other LECs. SPC$^{low}$MHC$^{high}$ LECs expressed the highest levels of the T cell coinhibitory molecule PD-L1, followed by AT2 cells then all other LECs (Fig. 2f, k). Multiciliated cells were high expressers of PD-L2 (Fig. 2f, l). ICAM1 was elevated on all LECs during active inflammation, but especially on the AT2 cells (Fig. 2j). Throughout the time-course, little to no CD80, ICOS-L, or VCAM1 were observed on any LEC (Supplementary Fig. 3c). The modulation of APC molecules on LECs on day 8 was microbe-induced, since the absence of infection did not incite such changes (Supplementary Fig. 4). These data show that all LECs express costimulatory molecules in a cell-specific fashion. Columnar LECs of the conducting airways (including club and multiciliated cells) have inducible MHC-II and constitutive expression of costimulators (CD40 and CD86, respectively), whereas the lamellar body-containing alveolar LECs (AT2 cells and SPC$^{low}$MHC$^{high}$ LECs) express constitutive MHC-II with PD-L1 (both of which increase during infection).

**LEC antigen presentation facilitates CD4$^+$ T$_{RM}$ cell niches around conducting airways**. We hypothesized that the anatomically segregated LEC antigen presentation pattern instructs the formation of CD4$^+$ T$_{RM}$ niches within the bronchovascular bundles along the airways[29]. To test this, we first tracked the temporal dynamics of CD4$^+$ T cells at timepoints coinciding with LEC antigen presentation using flow cytometry (as in Fig. 2d). To accurately discriminate between circulating versus lung CD4$^+$ T cells, we used an in vivo antibody labeling approach[30] and generated comprehensive snapshots of the dynamics of CD4$^+$ T

cell recruitment, expansion, and contraction leading to the formation of CD4$^+$ T$_{RM}$ niches (Fig. 3a and Supplementary Fig. 5a). One Sp19F infection instigated a modest but transient increase in CD4$^+$ T cells numbers by day 3 (Fig. 3a). We do not know the antigen specificity of these T cells, but at such an early time-point we suspect they were most likely not *Spn*-specific but were instead a mixture of naïve and unrelated cells that were recruited to the lung due to inflammation. Consistent with this, a single Sp19F infection did not induce lung CD4$^+$ T$_{RM}$ cells (Supplementary Fig. 5b). A second Sp19F infection at day 7; however, elicited rapid recruitment of all conventional CD4$^+$ T cell subsets including naïve T (T$_{Naive}$), central memory T (T$_{CM}$), effector memory T (T$_{EM}$), and effector T (T$_{eff}$) cells (Fig. 3a and Supplementary Fig. 5c). The CD4$^+$ T cell numbers peaked by day 10, with dominating subsets comprised of CD4$^+$ T$_{Naive}$, T$_{CM}$, and T$_{EM}$ cells along with activated and expanding T$_{eff}$ cells. At this point, ~17% of lung CD4$^+$ T$_{eff}$ cells expressed IL-2Rα (CD25), suggesting dependence on IL-2 signaling. Between days 10 and 24, the lung CD4$^+$ T cell compartment contracted until CD4$^+$ T$_{RM}$ cells and a small pool of patrolling CD4$^+$ T$_{EM}$ cells remained. CD4$^+$ T$_{eff}$ cells were the highest PD-1 expressers, with most surface PD-1 on day 10 (Fig. 3b, c). Interestingly, LEC PD-L1 expression spiked on day 8 (Fig. 2k) preceding the CD4$^+$ T$_{eff}$ PD-1 peak (Fig. 3c). Of note, only CD4$^+$ T$_{RM}$ cells were PD-1$^{high}$ in the resting but experienced lungs at day 35 (Fig. 3d).

We used immunofluorescence to localize CD4$^+$ cells during the establishment of CD4$^+$ T$_{RM}$ niches (Fig. 3e, f). The second Sp19F infection at day 7 elicited rapid recruitment of CD4$^+$ cells on days 8–10 in the loose interstitium of the bronchovascular bundles, surrounding the coordinated arterial and airway branching trees of the lower respiratory tract (Fig. 3e, f and Supplementary Fig. 5c). Resolution of infection between days 14 and 24 was associated with contraction of CD4$^+$ cells in parenchyma and bronchovascular bundles, except along the airway LECs where there appeared to be a stable CD4$^+$ cell niche (Fig. 3e, f). By day 35, 4 weeks after infection, all remaining CD4$^+$ cells were CD69$^+$ T$_{RM}$ cells in these resting lungs (Fig. 3a), and they localized exclusively around the conducting airways (Fig. 3e, f). These observations suggest that the airway niche promotes survival and/or maintenance of recruited CD4$^+$ T cells (perhaps via MHC-II plus the costimulatory CD40 and/or CD86 of the club and multiciliated cells) to form CD4$^+$ T$_{RM}$ cells, while the parenchymal niches instigate contraction of CD4$^+$ T cells (perhaps via LEC MHC-II plus coinhibitory molecules like PD-L1) during resolution of infection. If true, deletion of LEC MHC-II should disrupt the anatomical distribution of CD4$^+$ T$_{RM}$ niches within the lungs.

To test this, we generated CD4$^+$ T$_{RM}$ niches within MHC-II$^{fl/fl}$ and MHC-II$^{\Delta Epi}$ lungs and systematically quantified their anatomical distribution (Fig. 3g, h). The sustained absence of MHC-II was confirmed (Supplementary Fig. 5d). Consistent with the notion that LEC MHC-II facilitates the formation of CD4$^+$ T$_{RM}$ cell niches within bronchovascular bundles, MHC-II$^{\Delta Epi}$ lungs exhibited sparser CD4$^+$ T$_{RM}$ niches along the airway epithelium and within bronchovascular interstitium (Fig. 3h). However, loss of LEC MHC-II did not change the parenchyma CD4$^+$ T cell numbers, suggesting the involvement of other mechanisms in their contraction. For conducting airway LECs, MHC-II overlapped with CD40 better than CD86 (Fig. 2f–i), leading us to test whether the frequency of CD4$^+$ T$_{RM}$ cells in the lungs was dependent on CD40 signals offered by club cells in addition to other canonical CD40$^+$ cells. We examined this using the MR-1 antibody to block CD40-CD40L interaction during *Spn*-specific CD4$^+$ T$_{eff}$ cell recruitment (Fig. 3i). Consistent with the requirement for CD40 signals for the formation/maintenance of CD4$^+$ T$_{RM}$ cells, blockade of CD40-CD40L interactions significantly reduced lung CD4$^+$ T cell, CD4$^+$ T$_{RM}$ cell, and

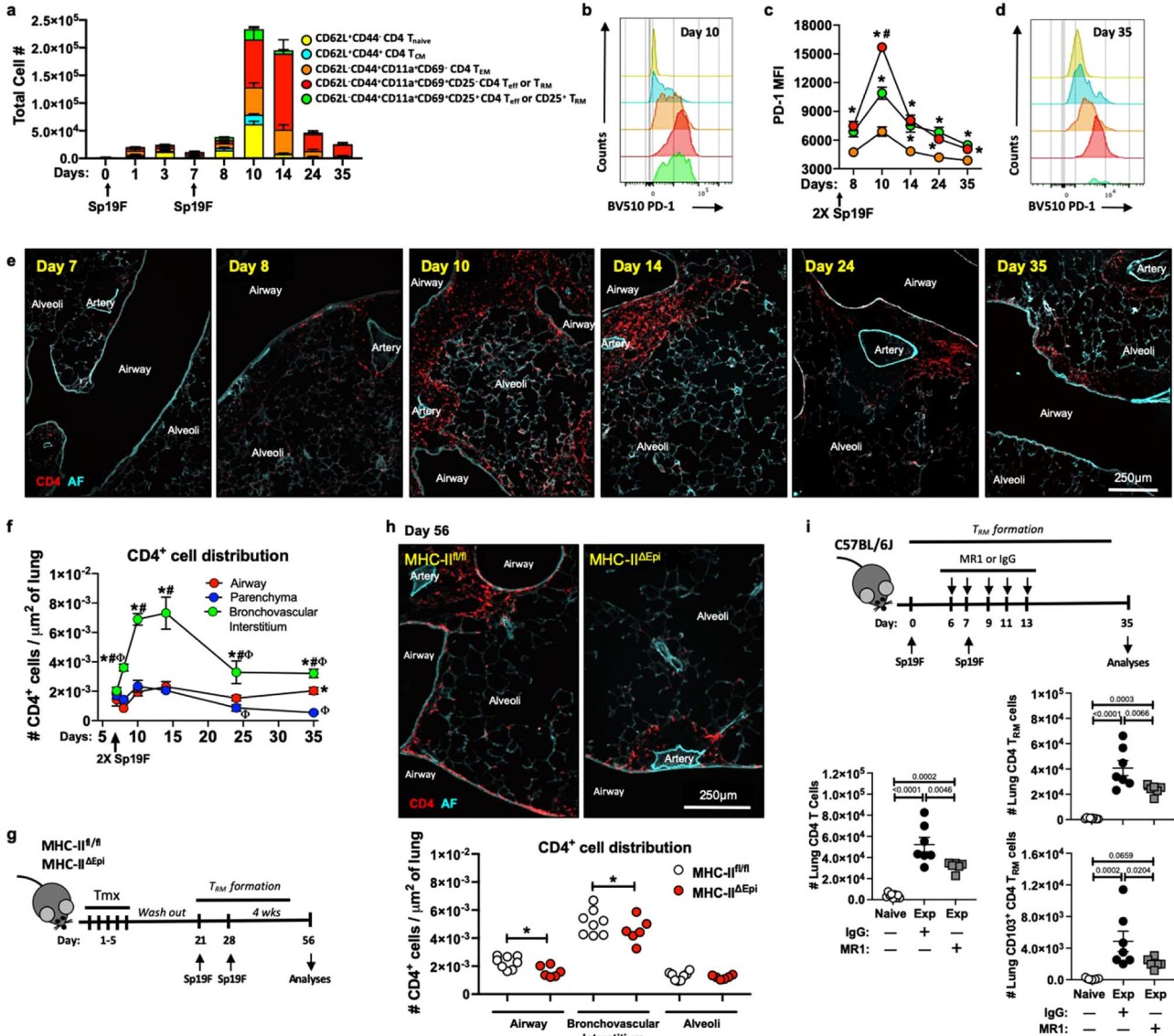

**Fig. 3 LEC MHC-II facilitates seeding of CD4$^+$ T$_{RM}$ niches around conducting airways. a** Numbers of lung (i.v.CD45.2$^-$) CD4$^+$ T cell subsets at designated timepoints, $n \geq 5$ mice/timepoint, two independent experiments, except day 10 ($n = 3$ mice). **b** Histogram for PD-1 on CD4$^+$ T cells at day 10. **c** PD-1 levels on CD4$^+$ T cells at designated timepoints, Two-way ANOVA with two-stage step-up method of Benjamini, Krieger, and Yekutieli to correct for multiple comparisons. FDR $q$ value: *$\leq$0.05 comparisons with T$_{EM}$, #$\leq$0.05 comparisons with CD25$^+$ T$_{eff}$ cells, $n = 6$ mice/timepoint, two experiments, except day 10 ($n = 3$ mice). **d** Histogram for PD-1 on CD4$^+$ T cells at day 35. **e** Representative immunofluorescent micrographs showing anatomical location of CD4$^+$ cells (*red*) at designated timepoints. Autofluorescence(AF) identifies lung structures. Data represents $n = 6$ mice/timepoint, except day 14 ($n = 5$ mice), two experiments. **f** Quantification of Fig. 3e, Two-way ANOVA with two-stage step-up method of Benjamini, Krieger, and Yekutieli to correct for multiple comparisons. FDR $q$ value: *$\leq$0.05 comparisons with parenchyma, #$\leq$0.05 comparisons with airway and $\Phi\leq$0.05 comparisons with day 8 within the niche, $n = 6$ mice/timepoint, except day 14 ($n = 5$ mice), two independent experiments. **g** Experimental timeline. **h** Representative immunofluorescent micrographs (*top*) and quantification (*bottom*) showing anatomical CD4$^+$ cell distribution within experienced MHC-II$^{fl/fl}$ and MHC-II$^{\Delta Epi}$ lungs at day 56. Data represents $n = 8$ MHC-II$^{fl/fl}$ and $n = 6$ MHC-II$^{\Delta Epi}$ mice, two experiments. One-Way ANOVA with two-stage step-up method of Benjamini, Krieger, and Yekutieli to correct for multiple comparisons. FDR $q$ value: *$\leq$0.05. **i** Timeline of CD40L blockade. Number of lung (i.v.CD45.2$^-$) CD4$^+$- T cells, T$_{RM}$ (CD11a$^{high}$CD69$^+$) cells, and CD103$^+$ T$_{RM}$ cells at day 35 in experienced mice with MR-1 (CD40L blockade) or IgG control, $n \geq 5$ mice, two independent experiments, One-Way ANOVA with Dunnett's multiple comparison test. All data are presented as mean ± SEM.

CD103$^+$ CD4$^+$ T$_{RM}$ cell numbers (Fig. 3i). Taken together, our results indicate that antigen presentation by LECs facilitates the formation of CD4$^+$ T$_{RM}$ niches around the conducting airways, but is dispensable for their exclusion from the lung parenchyma.

**CD4$^+$ T$_{RM}$ cells display multipotent phenotypes that are constrained by LEC MHC-II.** Residence of CD4$^+$ T$_{RM}$ cells within barrier tissues as frontline adaptive immune sentinels involves

inevitable and recurrent stimulations by cognate antigens over their lifespan. We next asked whether antigen presentation by LECs instructs CD4$^+$ T$_{RM}$ cell activities during repeated memory recall encounters. To test this, we generated CD4$^+$ T$_{RM}$ cells in MHC-II$^{\Delta Epi}$ and MHC-II$^{fl/fl}$ lungs as in Fig. 3g, followed by recurring exposures to a serotype-mismatched nonlethal strain of *Spn* serotype 23A (Sp23A) (Fig. 4a). Deletion of LEC MHC-II did not impede reactivation of CD4$^+$ T$_{RM}$ cells nor affected rapid clearance of *Spn* (Supplementary Fig. 6a–d) and as such, these

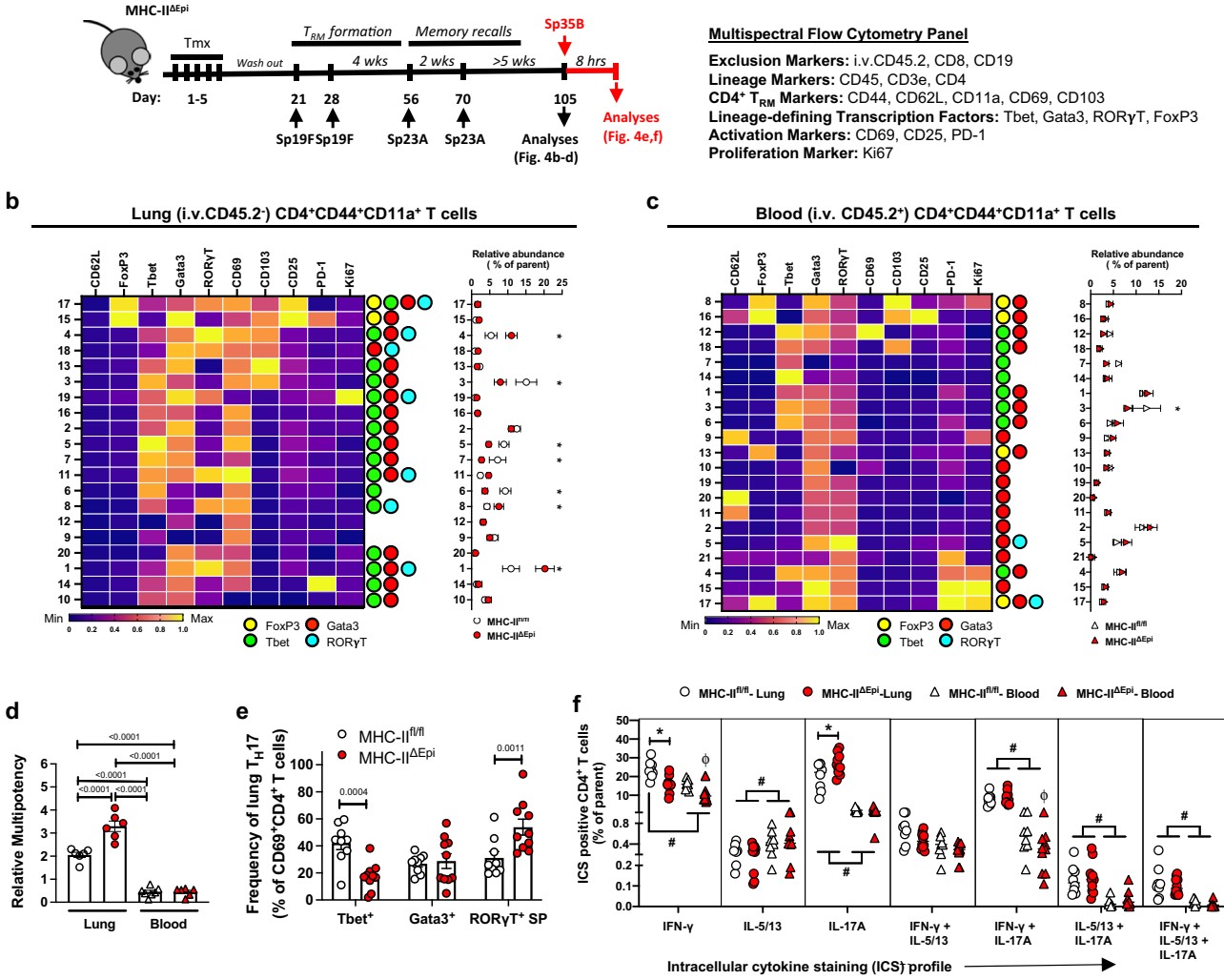

**Fig. 4 LEC MHC-II directs CD4⁺ T_RM cell activities. a** Schematic of experimental timeline and antibody panel used. **b Left:** Heat map depicting normalized expression levels of distinct molecules on lung (i.v.CD45.2⁻) memory (CD44⁺CD11a⁺) CD4⁺ T cells clusters on day 105. The lineage determining transcription factor (LDTF) status of each cluster is depicted. **Right:** Relative abundance of each cluster at day 105 in MHC-II^fl/fl and MHC-II^ΔEpi lungs. Two-way ANOVA with two-stage step-up method of Benjamini, Krieger, and Yekutieli to correct for multiple comparisons. FDR p value: *≤ 0.05. **c Left:** Heat map depicting normalized expression levels of distinct molecules on blood intravascular (i.v.CD45.2⁺) memory (CD44⁺CD11a⁺) CD4⁺ T cell clusters on day 105. LDTF status of each cluster is depicted. **Right:** Relative abundance of each cluster at day 105 in MHC-II^fl/fl and MHC-II^ΔEpi lungs. Two-way ANOVA with two-stage step-up method of Benjamini, Krieger and Yekutieli to correct for multiple comparisons. FDR p value: *≤ 0.05. **d** Relative multipotency indices for memory (CD44⁺CD11a⁺) CD4⁺ T cells in lung (*circle*) and blood (*triangles*) of MHC-II^fl/fl (*white*) and MHC-II^ΔEpi mice (*red*) at day 105, One-way ANOVA with Holm-Sidak's multiple comparison test. **e** Frequency of activated (CD69⁺) Tbet⁺, Gata3⁺ or RORγT single-positive (SP) T_H17 cells in MHC-II^fl/fl and MHC-II^ΔEpi lungs 8hpi Sp35B, Two-Way ANOVA with Fisher's LSD test. For all experiments, positivity for a LDTF was determined using cutoffs identified from clusters negative for that LDTF. **f** Intracellular cytokine staining (ICS) profile of lung (i.v.CD45.2⁻) and blood (i.v.CD45.2⁺) CD4⁺ T cells isolated from lungs on day 105 and stimulated with PMA/Ionomycin ex vivo, two-way ANOVA with Geisser-Greenhouse correction and two-stage step-up method of Benjamini, Krieger, and Yekutieli to correct for multiple comparisons. p value: #≤ 0.05 comparison between lung and blood; *≤ 0.05 genotype-dependent comparison within lungs; Φ ≤ 0.05 genotype-dependent comparison within blood. All data have n ≥ 6 mice, two independent experiments. All data are presented as mean ± SEM.

infections mimic homeostatic nonlethal inhalation of microbes to which adult lungs would already possess CD4⁺ T_RM cells. Measurements of airway cellularity revealed no ongoing inflammation as evidenced by <1% neutrophil content of the bronchoalveolar lavage (BAL) fluid on day 105 (Supplementary Fig. 6e). Sustained deletion of LEC MHC-II was confirmed (Supplementary Fig. 6f).

To examine CD4⁺ T_RM phenotypes in the absence of LEC MHC-II, we used a 21-parameter high-dimensional MSFC panel including antibodies to CD4⁺ T_RM markers, T cell lineage-defining transcription factors (LDTFs), and markers of activation and proliferation (Fig. 4a). Analysis of total lung CD4⁺ T cell and

CD4⁺ T_RM cell numbers on day 105 revealed comparable frequencies between genotypes (Supplementary Fig. 6g). Next, we used Phenograph to identify distinct clusters of CD4⁺ T cells based on unique expression profiles, and quantified their relative abundance in genotype-dependent fashion. Phenograph clustering of concatenated lung (i.v.CD45.2⁻) CD44⁺CD11a⁺ CD4⁺ T cells identified 21 distinct memory CD4⁺ T cell clusters (Supplementary Fig. 7a), none being unique to either genotype and all being negative for CD62L. For further analyses, we focused on the 20 clusters at >0.5% frequency (Fig. 4b). Remarkably, 17 of the 20 memory CD4⁺ T cell clusters

co-expressed ≥2 LDTFs (cluster 17 was positive for all LDTFs tested) (Fig. 4b). Among these, clusters 10 and 19 were CD69$^-$ and represented $T_{EM}$ cells while the remaining 15 clusters encompassed $T_{RM}$ cells comprised of 2 CD103$^+$ $T_{reg}$-[31], 1 $T_H1/17$-, 1 $T_H2/17$-, 8 $T_H1/2$-, and 3 $T_H1/2/17$- lung $T_{RM}$ clusters at homeostasis. Thus, our results highlight the presence of complex lineage plasticity and multipotency within lung CD4$^+$ $T_{RM}$ cells. Of note, clusters 9 and 12 were negative for all tested LDTFs; however, positivity for other LDTFs not included in our panel, like BCL6, cannot be excluded.

Quantification of relative frequencies revealed significant perturbations in 8 of the 20 memory CD4$^+$ T cell clusters due to deletion of LEC MHC-II (Fig. 4b). While MHC-II$^{\Delta Epi}$ mice exhibited significant expansion of $T_H1/2/17$ $T_{RM}$ clusters 1 and 4, and the $T_H1/17$ $T_{RM}$ cluster 8, they revealed a significant contraction in 3 $T_H1/2$ $T_{RM}$ clusters (3, 5, and 7) and the $T_H1$ $T_{RM}$ cluster 6 (Fig. 4b). Of note, all CD4$^+$ $T_{RM}$ clusters enlarged in MHC-II$^{\Delta Epi}$ mice were Ki67$^-$, suggesting they were not the result of uninhibited proliferation at tissue homeostasis. Furthermore, neither of the $T_{reg}$ $T_{RM}$ cell clusters were affected by LEC MHC-II deletion, implying that the observed effects were probably independent of $T_{reg}$-suppressive functions. Our findings, therefore suggest that LEC MHC-II imposes inhibitory constraints on CD4$^+$ $T_{RM}$ cell plasticity and multipotency (defined as coexpression of ≥2 LDTFs).

Next, we investigated whether deletion of LEC MHC-II induces perturbations in the circulating CD4$^+$ T cell compartment by performing Phenograph clustering on the intravascular (i.-v.CD45.2$^+$) CD44$^+$CD11a$^+$ CD4$^+$ T cells collected from the lungs of the same mice. We identified 21 distinct clusters (Supplementary Fig. 7b), none being unique to either genotype. In stark contrast to the lungs, only 11 of the 21 blood memory CD4$^+$ T cell clusters exhibited multipotency (Fig. 4c) with 10 of the 21 clusters being positive for only one of these LDTFs (Fig. 4c). Comparisons of relative frequencies for multipotency revealed stark differences in the permitted degree of plasticity conferred to memory CD4$^+$ T cells based on: (i) the status of LEC MHC-II and (ii) the anatomical site of T cell residence. For example, while 81% of MHC-II$^{fl/fl}$ lung CD4$^+$ $T_{RM}$ cells exhibited multipotency (of which 20% were $T_H1/2/17$), a significantly higher proportion (88%) of MHC-II$^{\Delta Epi}$ lung CD4$^+$ $T_{RM}$ cells coexpressed ≥2 LDTFs (of which 38% were $T_H1/2/17$) (Supplementary Fig. 7c). Irrespective of their LEC MHC-II status, only ~61% of all blood memory CD4$^+$ T cells displayed such plasticity (none of which were $T_H1/2/17$) (Supplementary Fig. 7c). Comparing phenotypes of extravascular (tissue) and intravascular (circulating) CD4$^+$ T cells in the lungs of MHC-II$^{fl/fl}$ and MHC-II$^{\Delta Epi}$ mice revealed that the tissue cells had greater complexity than the circulating pools (evidenced by Tbet$^+$Gata3$^+$RORγT$^+$ triple-positive CD4$^+$ T cells corresponding to $T_H1/2/17$ skewing) (Supplementary Fig. 7c) with consistently high multipotency (Fig. 4d) that was exacerbated by the loss of LEC MHC-II (Fig. 4d and Supplementary Fig. 7c). Our results, thus (i) ascertain antigen presentation by LECs as a critical regulator of CD4$^+$ $T_{RM}$ cell plasticity and (ii) emphasize the existence of distinct selective pressures against memory CD4$^+$ T cell plasticity in circulation versus barrier tissues of the same immunocompetent host.

LEC MHC-II may govern CD4$^+$ $T_{RM}$ cell plasticity by two ways: (i) by constraining plasticity of pre-existing $T_H17$-specified CD4$^+$ $T_{RM}$ cells during memory recalls or instead (ii) by regulating the development of multipotent CD4$^+$ $T_{RM}$ cells from $T_{eff}$ cells[29]. To test these possibilities, we first generated CD4$^+$ $T_{RM}$ cells in C57BL/6J, MHC-II$^{fl/fl}$, and MHC-II$^{\Delta Epi}$ mice and sought to examine if newly established CD4$^+$ $T_{RM}$ cells were exclusively $T_H17$-specified (Supplementary Fig. 8a). Consistent with the notion that CD4$^+$ $T_{RM}$ cells are not solely $T_H17$-specified, we observed heterogeneity

and plasticity in CD4$^+$ $T_{RM}$ cells even at day 56 (Supplementary Fig. 8b, c). Furthermore, we detected modest but significant expansion of $T_H1/2/17$ CD4$^+$ $T_{RM}$ cells in MHC-II$^{\Delta Epi}$ mice even without subsequent Sp23A memory recalls (Supplementary Fig. 8c–e), suggesting that CD4$^+$ $T_{RM}$ cells are innately plastic and LEC MHC-II might regulate multipotency development from tissue-infiltrating $T_{eff}$ cells. To confirm this, we generated CD4$^+$ $T_{RM}$ cells in MHC-II$^{fl/fl}$ and MHC-II$^{\Delta Epi}$ mice before deletion of LEC MHC-II, and then provided Sp23A memory recall exposures in presence of the lymph node egress inhibitor FTY720 (Supplementary Fig. 9a). This denied CD4$^+$ T cells access to the systemic circulation (Supplementary Fig. 9b–d) so that only pre-established lung CD4$^+$ $T_{RM}$ cells could contribute to the memory recall responses and plasticity depending on LEC MHC-II status. No differences in CD4$^+$ $T_{RM}$ cell phenotypes between MHC-II$^{fl/fl}$ and MHC-II$^{\Delta Epi}$ lungs were observed due to LEC MHC-II deletion (Supplementary Fig. 9e–h). Together, these observations suggest that LEC MHC-II does not govern phenotypes of established $T_{RM}$ cells but rather constrains the multipotency of CD4$^+$ $T_{RM}$ cells developing from $T_{eff}$ cells during primary and memory recall encounters.

**LEC MHC-II directs skewing of lung CD4$^+$ $T_{RM}$ cell responses.** We next investigated the activity of the multipotent $T_{RM}$ cells on restimulation with a nonlethal mismatched *Spn* serotype (Sp35B) in order to elicit a heterotypic response. We profiled lung CD4$^+$ T cells 8 hours after infection (Fig. 4a), when *Spn*-specific $T_{RM}$ cells already bolster neutrophil recruitment[11,19]. Concatenated lung CD62L$^-$CD44$^+$CD11a$^+$ CD4$^+$ T cells included 26 distinct clusters (Supplementary Fig. 10a), none being unique to either genotype. Unlike resting $T_{RM}$ cells, no CD69$^+$ clusters in the infected lung were triple positive for Tbet, Gata3, and RORγT (Supplementary Fig. 10b, c), suggesting a rapid transition away from extensive multipotency towards more limited phenotypes with one or two dominant LDTF.

Quantification of relative frequencies revealed significant perturbations in 4 of 15 CD69$^+$ T cell clusters (Supplementary Fig. 10b) within MHC-II$^{\Delta Epi}$ lungs. Of these, $T_H17$ clusters 3 and 8 and $T_H1$ cluster 7 were enriched in MHC-II$^{\Delta Epi}$ lungs, while the $T_H1/17$ cluster 6 was reduced, compared to MHC-II$^{fl/fl}$ littermates. Quantification of cumulative frequencies identified a significant expansion of $T_H17$ cells with a concomitant reduction of the less-specified $T_H1/17$ cells within the MHC-II$^{\Delta Epi}$ lungs (Supplementary Fig. 10c). Cumulative frequencies of each individual transcription factor within RORγT$^+$ T cells further suggested a bias towards $T_H17$ and away from $T_H1$ in the absence of LEC MHC-II (Fig. 4e), based on LDTF content. This was further supported by the strong negative correlation of Tbet$^+$ $T_H17$ cells with RORγT$^+$ single-positive (SP) $T_H17$ cells on memory recall (Supplementary Fig. 10d). Of note, these differences were independent of blood CD4$^+$ T cell involvement since no genotype-dependent differences were observed in the pulmonary intravascular memory cells (i.v.CD45.2$^+$CD62L$^-$CD44$^+$CD11a$^+$CD4$^+$) of the same mice (Supplementary Fig. 11). Thus, based on LDTFs, the absence of LEC MHC-II exaggerates lung $T_H17$ cell responses, at the cost of $T_H1$ activity.

To define T cell lineages by cytokine expression, we used intracellular cytokine staining with PMA/Ionomycin stimulated lung and blood CD4$^+$ T cells from experienced MHC-II$^{\Delta Epi}$ and MHC-II$^{fl/fl}$ mice. Across conditions, the majority of these CD4$^+$ T cells were either IFN-γ$^+$ or IL-17A$^+$, followed by IFN-γ$^+$IL-17A$^+$ dual expressor cells (Fig. 4f). Only very few cells expressed IL-5 or IL-13. Consistent with the absence of MHC-II on LECs leading to reprogrammed $T_{RM}$ cells, lung CD4$^+$ T cells from MHC-II$^{\Delta Epi}$ mice expressed increased IL-17A and decreased

IFN-γ when stimulated (Fig. 4f). The latter finding was recapitulated by antigen-specific ex vivo stimulation of CD4[+] T cells (Supplementary Fig. 12). Very consistently, $T_H1$ activity was blunted in CD4[+] T cells from lungs devoid of LEC MHC-II.

**The absence of LEC MHC-II dysregulates barrier immune landscape.** CD4[+] $T_{RM}$ cells confer microbial clearance by accelerating innate immunity[1,2]. We examined if LEC MHC-II and its regulation of CD4[+] $T_{RM}$ cell activities instructs innate immune cellular responses to memory recall infections. While both the MHC-II[fl/fl] and MHC-II[ΔEpi] lungs exhibited comparable Sp35B clearance (Fig. 5a) and BAL cells (Fig. 5b), quantification of extravascular myeloid cells in the infected lung using flow cytometry revealed significant perturbations in the lung immune landscape of MHC-II[ΔEpi] mice within 8 hours of heterotypic infection (Fig. 5c, and Supplementary Fig. 13, 14). MHC-II[ΔEpi] lungs possessed higher frequencies of eosinophils and tissue-resident alveolar macrophages, but they exhibited reduced accumulation of multiple monocyte-related cells including Ly6C[+] inflammatory monocytes, monocyte-derived interstitial macrophages, and monocyte-derived DCs, in addition to reduced CD11b[+] conventional DCs and plasmacytoid DCs (Fig. 5c). These results suggest that deletion of LEC MHC-II has far-reaching consequences on barrier immunity. Of note, these changes were downstream of CD4[+] $T_{RM}$ cell reactivation, since uninfected lungs exhibited no inflammation (Supplementary Fig. 6e) or decrements in monocyte-related cell compartments (Supplementary Fig. 15a, b), although eosinophil enrichment was apparent (Supplementary Fig. 15a, b). To connect CD4[+] T cell clusters to the observed dysregulation in barrier immune landscape on memory recall, we performed correlation analyses for each CD69[+]CD4[+] T cell lineage and all the myeloid cell changes from the same mice (Fig. 5d). Consistent with the notion that RORγT[+] $T_{reg}$ cells suppress $T_H17$ responses[32], RORγT[+] $T_{reg}$ cells inversely correlated with neutrophil responses. Furthermore, consistent with Tbet promoting monocyte influx[33], $T_H1/17$ exhibited direct correlation with Ly6C[+] monocytes and monocyte-derived CD11b[+] DCs. Interestingly, we observed strong and direct correlations between eosinophils and $T_H17$ cells and lung neutrophils with LDTF-negative (TN) CD4[+] $T_H$ cells, suggesting possible T cell influences on myeloid cells.

Reduced lung monocytes and persistently high eosinophils in the MHC-II[ΔEpi] lungs evoke features reported in patients with checkpoint blockade immunotherapy (CBI)[34–37] and checkpoint inhibitor pneumonitis (CIP)[38]. We probed the leukocyte frequencies within the blood fraction (i.v.CD45.2[+]) of MHC-II[fl/fl] and MHC-II[ΔEpi] lungs for features proposed as robust clinical biomarkers for CBI[34–37,39,40]. Consistently, blood fraction of MHC-II[ΔEpi] lungs phenocopied features of CBI as manifested by presence of elevated eosinophil frequencies, reduced monocyte frequencies, elevated lymphocyte to monocyte ratios, elevated eosinophil-to-monocyte ratios, and elevated eosinophil-to-lymphocyte ratios during infection (Fig. 5e, f and Supplementary Fig. 15c, d). Our results thus demonstrate that memory recall of CD4[+] $T_{RM}$ cells expanded in absence of LEC MHC-II triggers dysregulated barrier immunity phenocopying features of CBI, emphasizing the complex (and incompletely understood) intra- and interleukocyte interactions mediating barrier immunity.

**LEC MHC-II functions in a post-translational coupling with PD-L1.** The similarities of immune dysregulations in infected MHC-II[ΔEpi] mice and patients with CBI and CIP led us to further investigate checkpoint molecules on LECs. LEC MHC-II correlated strongly with LEC PD-L1 throughout all timepoints and across all LEC-types (Fig. 6a). Higher PD-L1 levels on MHC-II[high] LECs were also confirmed in human lungs, which had similar positive

correlations between the 2 signals (Fig. 6b). When we examined surface PD-L1 expression on LECs of MHC-II[ΔEpi] mice, we unexpectedly observed loss of this immune checkpoint on alveolar LECs during homeostasis and pneumonia (Fig. 6c and Supplementary Fig. 16a). This coupling also extended to other lung challenges (Supplementary Fig. 17), suggesting an evolutionarily conserved coupling between MHC-II and PD-L1 on LECs leading to an unanticipated LEC-specific interruption of checkpoint signaling. We aimed to elucidate the mechanism underlying the loss of PD-L1 on MHC-II[ΔEpi] LECs. There was a comparable expression of *Pd-l1* mRNA in MHC-II[ΔEpi] and MHC-II[fl/fl] LECs (Supplementary Fig. 16b), suggesting post-transcriptional effects. We posited that PD-L1 depended on LEC MHC-II for surface localization, and tested this using a monoclonal antibody that binds to PD-L1 extracellular domains[41]. Consistent with our hypothesis, while most PD-L1 of MHC-II[fl/fl] alveolar LECs was surface exposed, almost all PD-L1 within MHC-II[ΔEpi] alveolar LECs was trapped intracellularly (Fig. 6d). In addition to failed PD-L1 delivery to the cell surface, distal LECs exhibited decreased content of intracellular PD-L1 (Supplementary Fig. 16c), consistent with PD-L1 instability and degradation in MHC-II[ΔEpi] LECs. These data suggest that surface presentation of the checkpoint molecule PD-L1 is entirely dependent upon the expression levels of MHC-II in LECs. This absence of LEC PD-L1 may contribute to phenotypes resulting from LEC MHC-II loss, as evidenced in PD-1 deficient mice[42] with relevant pneumonia history (Fig. 6e) showing greater memory CD4[+] $T_{RM}$ cell plasticity (Fig. 6f and Supplementary Fig. 18) and multipotency (Fig. 6g) in their lungs but not their blood, akin to MHC-II[ΔEpi] mice. Taken together, MHC-II on LECs emerges as having two-pronged effects on immunity, directing adaptive immunity via antigen presentation and regulating adaptive immunity via PD-L1.

## Discussion

Herein, we show that diverse LECs, including a distinct SPC[low]MHC[high] LEC, function as APCs to CD4[+] T cells in anatomically segregated fashions (Supplementary Figure 19a). We discover that LEC MHC-II is dynamically regulated on all LECs in conjunction with CD4[+] T cell recruitment and activation (Supplementary Figure 19b) and that LEC MHC-II imposes constraints on aberrant expansion and plasticity of CD4[+] $T_{RM}$ cells (Supplementary Figure 19c) via the PD-L1:PD-1 signaling axis wherein epithelial PD-L1 depends on MHC-II. Further, we show that disruption of LEC MHC-II has significant consequences on the barrier immune landscape, thus, establishing LEC-CD4[+] $T_{RM}$ cell immune synapses as core elements of barrier immunity. Epithelial antigen presentation, thus, plays a pivotal role in localizing and directing CD4[+] $T_{RM}$ cell immunity.

MHC-II expression is high on AT2 cells and SPC[low]MHC[high] LECs. SPC[low]MHC[high] LECs possess features of traditional club cells and MHC-I[high] club-like stem cells[23], but they resemble AT2 cells in their high MHC-II, lamellar bodies, and abundance within the lung. They may represent an AT2 cell subset. Given their overlaps with stem cells of the lung[23,43], and reported connections of MHC-II to epithelial stem cell biology in other barrier tissues[15,17], the immunological synapse may be a component of tissue repair and SPC[low]MHC[high] LECs may contribute to resolution, repair, or regeneration. If and how SPC[low]MHC[high] LECs compare to Wnt-active AT2-stem cells[44,45] and AT2-signaling cells[46] remains to be determined. CD4[+] $T_{RM}$ cells can regulate stem cell renewal, proliferation, differentiation fate, and tissue repair in the skin and the gut[6,15]. Similar roles may apply for MHC-II in regulating lung stem cell functions[47].

Epithelial MHC-II has been connected to regeneration, but its immune significance is less established. Our results identify antigen presentation as a pan-epithelial feature with major roles

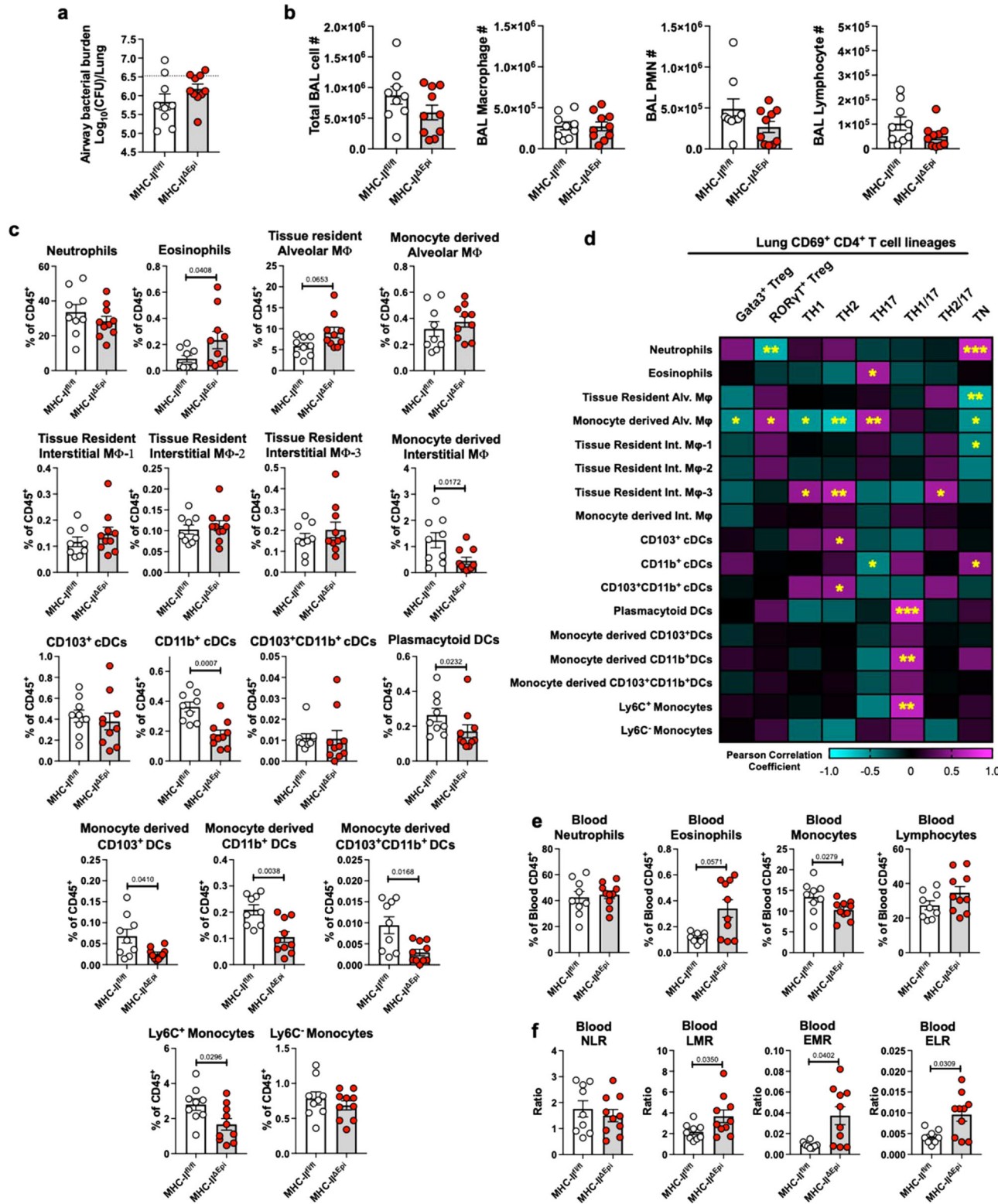

in lung and CD4$^+$ T$_{RM}$ biology. MHC-II on LECs was envisioned as restricted to AT2 cells[14,48], but now is expanded to SPC$^{low}$MHC$^{high}$ LECs and to club cells and multiciliated cells during inflammation. Possible immune roles for epithelial MHC-II were controversial and speculative[20,49–51]. Here, we provide in vivo evidence of functional significance whereby deletion of LEC MHC-II led to decreased seeding and aberrant phenotypes of CD4$^+$ T$_{RM}$ cells around conducting airways, with diminished recruitment of monocyte-derived cells during subsequent

infections. Thus, LEC MHC-II is a critical tissue-localized cue for resident memory cells.

Our results identified a surprising degree of transcriptional plasticity and multipotency within CD4$^+$ T$_{RM}$ cells, more than previously appreciated[6,19,29]. Adult humans daily inhale ~11,000 liters of air with its contaminants, regularly microaspirate, and experience frequent respiratory infections, a tremendous environmental exposure for lung memory cells. Factors constraining CD4$^+$ T$_{RM}$ cell plasticity driven by this environmental pressure

**Fig. 5 MHC-II$^{\Delta Epi}$ mice exhibit dysregulated barrier immunity. a** Airway Sp35B burden 8hpi. **b** Total bronchoalveolar lavage (BAL) cellularity, macrophages, neutrophils (PMNs) and lymphocytes 8hpi Sp35B. **c**. Frequencies of major lung (i.v. CD45.2$^-$) myeloid cell populations identified 8hpi Sp35B, two-tailed Mann–Whitney test. Note, the mice used in these experiments belonged to both sexes and were of various sizes at the experiment endpoint (~6-8 months old at euthanasia). This was reflected in differences in lung sizes and hence, the frequency data herein and absolute numbers in Supplementary Fig. 14 are both used to draw conclusions. **d** Heat map representation of two-tailed Pearson correlation analyses of frequencies of the lung (i.v. CD45.2$^-$) activate (CD69$^+$) CD4$^+$ T cell lineages with frequencies of major myeloid cells identified in Fig. 5c in lungs 8hpi Sp35B. p-value: *$\leq$ 0.05, **$\leq$ 0.01, ***$\leq$ 0.001. **e** Frequencies of intravascular (i.v.CD45.2$^+$) immune cells depicted as fraction of i.v.CD45.2$^+$ cells 8hpi Sp35B. two-tailed Mann–Whitney test. **f** Blood neutrophil-to-lymphocyte ratio (NLR), lymphocyte-to-monocyte ratio (LMR), eosinophil-to-monocyte ratio (EMR) and eosinophil-to-lymphocyte ratio (ELR) in i.v.CD45.2$^+$ fraction of lungs 8hpi Sp35B. two-tailed Mann–Whitney test. All experiments have $n \geq 8$ mice, 2 independent experiments. All data are presented as mean ± SEM.

were unknown. Our studies reveal 3 major findings. (i) CD4$^+$ T$_{RM}$ cells possess extensive flexibility, wherein many T$_{RM}$ cells co-express $\geq$2 classically antagonistic LDTFs. Such plasticity within T$_H$17 effector cells has been reported[52] but our work extends these paradigms to T$_{RM}$ cells and suggests a stem-cell-like uncommitted property for otherwise "primed" T$_{RM}$ cells that might allow flexibility to choose effector pathways suited to the nature of perceived threats[3]. Epigenetic accessibility of LDTFs to target genes defines T cell responses, and future studies using single-cell epigenetic and transcriptomic profiling of multipotent T$_{RM}$ cells are warranted[5,53]. Consistent with T$_H$17 effector cells, we observed a tendency for T$_H$17 T$_{RM}$ cell clusters to acquire a T$_H$2 Gata3 signal at homeostasis[52], but this was not sustained at the cytokine level post-restimulation. Whether Gata3 expression reflects bivalent epigenetic marks adjacent to *Gata3* gene within T$_{RM}$ cells[53] or indicates a poised T$_H$2 transcriptome that would translate into T$_H$2 cytokines upon licensing by alarmins such as IL-18 and IL-33 (which are precluded in our 8-h challenge model with *Spn*)[6] needs further investigation. (ii) LECs impose constraints on the aberrant expansion of T$_{RM}$ cells co-expressing Tbet, Gata3, and RORγT via PD-L1:PD-1 signaling axis. Our study establishes PD-1 as a checkpoint molecule that restrains CD4$^+$ T cell phenotypic boundaries in peripheral non-lymphoid tissues. In doing so, we distinguish these restraints as distinct from those imposed by CTLA-4 during initial priming within the lymph nodes in that they curtail expansion, but not *de novo* generation, of multipotent T$_{RM}$ cell clusters[54]. Given that MHC-II and PD-L1 expression in LECs is anatomically segregated and enriched in alveolar LECs, the physical locations of the most multipotent T$_{RM}$ cells within the lung becomes an important next question. (iii) Our findings show that the lungs provide a permissive environment for memory CD4$^+$ T cell plasticity compared to the blood. We propose that this may reflect distinct selective pressures in barrier tissues where antigenic encounters are more diverse in terms of quantity, quality, and frequency.

Deletion of LEC MHC-II altered the barrier immune landscape. Reactivated CD4$^+$ T$_{RM}$ cells expanded in the absence of LEC MHC-II show an intrinsic bias to deviate away from T$_H$1 responses, precipitating consequences on host immunity that include increased alveolar macrophages and eosinophils accompanied by reduced monocytes, monocyte-derived interstitial macrophages, monocyte-derived DCs, CD11b$^+$ cDCs, and pDCs within MHC-II$^{\Delta Epi}$ lungs. While molecular mechanisms are likely multifactorial, it is reasonable to posit that reduced T$_H$1 responses tilted the barrier immune landscape towards weaker monocyte recruitment and more eosinophilia[33,55]. This remodeled immunity phenocopies biomarkers of successful CBI[34–37,39,40] and is in line with higher incidences of CIP in PD-1/PD-L1 CBI patients[56]. Our observations thus suggest that PD-1/PD-L1 CBI may interfere with epithelial constraints on CD4$^+$ T$_{RM}$ cells and unleash unrestrained T$_{RM}$ cell activation leading to immunopathologies like CIP, colitis, IBD, and skin inflammation, all of which may involve CD4$^+$ T cell plasticity[7–9,56,57]. PD-L1 blockade results in

unrestrained immunopathology of the lungs in response to viral antigens[58], and a similar axis for restraining CD8$^+$ T$_{RM}$ cell-driven pathology in the human pancreas is exploited by PD-L1-expressing pancreatic macrophages[59]. The exact biological role for the exhaustion marker PD-1 on CD4$^+$ T$_{RM}$ cells that are otherwise highly responsive to antigenic restimulation is incompletely understood[1]. Our results suggest that T$_{RM}$ cell PD-1 may act akin to a molecular switch that an organ like the lung can exploit to limit the plasticity of T$_{RM}$ cells arising there.

Our findings have implications that pervade fields beyond barrier epithelial and CD4$^+$ T$_{RM}$ cell biology. High levels of MHC-II on AT2 cells[14], keratinocytes[17], and Lgr5$^+$ intestinal stem cells[15] connect antigen presentation to diverse tumor-initiating stem cells (TISCs)[60–62]. While PD-L1 expression on Lgr5$^+$ ISCs is uncertain, keratinocytes, like AT2 cells and SPC$^{low}$MHC$^{high}$ LECs, do express PD-L1[63]. Given our observations, it is tempting to posit that MHC-I, MHC-II, and PD-L1 co-expression on TISCs may jointly be responsible for evasion of early antitumor T cell immuno-surveillance during tumorigenesis. Furthermore, high MHC-II and PD-L1 levels might also provide a tumor-conducive immunosuppressive microenvironment wherein expansion of antitumor CD4$^+$ T cells would be restricted[64]. Indeed, murine lungs deficient in MHC-II within non-hematopoietic cells are resistant to metastatic tumorigenesis[65]. Studies investigating the tumorigenic potential of SPC$^{low}$MHC$^{high}$ LECs and elucidating the role of MHC-II in epithelial stem cells are merited. Detailed understanding of stem cell antigen presentation will help guide the development of innovative immunomodulatory agents and inform development of molecular targets that could be valuable biomarkers for early cancer detection and immunotherapy outcomes, such as MHC-II for melanoma[66].

Taken together, our study identifies antigen presentation as a pan-epithelial feature that exhibits division of labor in anatomically segregated and temporally dynamic fashions within mammalian lungs. We identify epithelial antigen presentation as a tissue-resident regulator of CD4$^+$ T$_{RM}$ cell locations, plasticity, and activities, and implicate disruption of epithelial restraints on CD4$^+$ T$_{RM}$ cells as underlying checkpoint blockade induced immune dysregulation. Distinct epithelial cells in other barrier tissues may have similar rheostat functions in T$_{RM}$ cell immunity. Epithelial MHC is emerging as relevant to stem cell biology, cancer, and immunomodulatory therapy. Thus, considering epithelial antigen presentation may inform future strategies to curb immunopathology during chronic diseases and checkpoint blockade.

## Methods
**Mice**. C57BL/6J (Stock# 000664), OT-II TCR transgenic (B6.Cg-Tg(TcraTcrb) 425Cbn/J) (Stock # 004194), and B6.Cg-*Pdcd1*$^{tm1.1Shr}$/J (PD-1$^{-/-}$, Stock #028276)[42] mice were obtained from The Jackson laboratories (USA) at 6 weeks old age, all on the C57BL/6 background. SPC-GFP mice[20] were back-crossed more than 15 generations onto C57BL/6 and bred in Boston University animal facilities. Mice for targeted deletion of MHC-II on LECs were generated in-house at BU-ABSL2 facility. Briefly, male B6.129X1-H2-Ab1$^{tm1Koni}$/J (Stock# 013181; C57BL/6 background) and female Nkx2-1$^{tm1.1(cre/ERT2)Zjh}$/J mice (Stock# 014552; B6.129SF2 background) were obtained from The Jackson Laboratories (USA) and

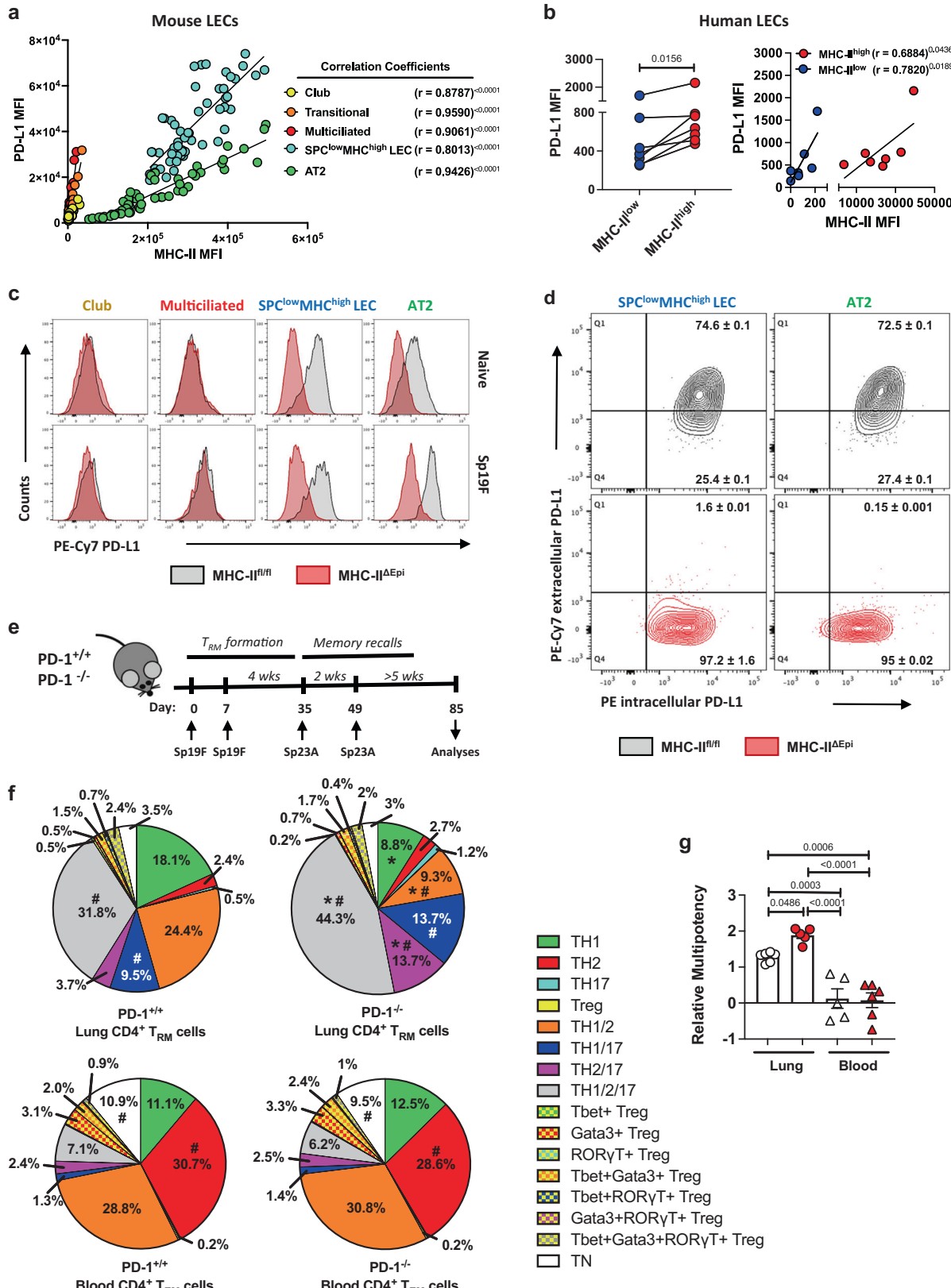

crossed to create Nkx2.1$^{creERT2}$H2-Ab1$^{fl/fl}$ mice. Both founder mice lack I-E due to mutation in H2-Ea locus allowing deletion of H2-Ab1 to abrogate MHC-II entirely. Nkx2-1 is expressed in all LECs and some cells of telencephalon, thyroid, and pituitary[67–69] and Nkx2-1-driven Cre efficiently targets all LECs[70]. All breeders were homozygous floxed for H2-Ab1. For experiments, all mice were homozygous for loxP sites flanking H2-Ab1 exon1 and were identified as Cre-positive or negative based on the presence or absence of Nkx2.1$^{cre/ERT2}$. Cage and littermate

controls of both sexes were used for the studies. BASC Reporter mice including the BASC viewer and the BASC v-race mice were on a mixed hybrid C57BL/6 x 129/SV background[21]. All mice were housed in specific pathogen-free environment on a 12-h light-dark cycle at 30–70%, humidity and temperature of 20–26 °C, with ad libitum access to standard chow and water. for all experiments, 7–14-week-old mice were used. Mice were euthanized using isoflurane overdose and death confirmed using pneumothorax before organ collections. Animal procedures were

**Fig. 6 LEC PD-L1 functions in a post-translational coupling with MHC-II. a** Scatterplot correlating MHC-II and PD-L1 on LECs across nine timepoints designated in Fig. 2d ($n = 54$ mice, two independent experiments). Two-tailed Pearson's correlation coefficient (r) and p values denoted. **b** PD-L1 levels on human LECs ($n = 7$). Two-tailed Wilcoxon matched-pairs signed-rank test. Scatterplot correlating MHC-II and PD-L1 on human LECs ($n = 7$). One-tailed Pearson's correlation coefficient (r) and p values denoted. **c** Histogram for surface PD-L1 levels on LECs from naïve and Sp19F-infected MHC-II$^{fl/fl}$ and MHC-II$^{\Delta Epi}$ mice 48hpi. $n \geq 4$ mice, two independent experiments. **d** Contour plots for subcellular localization of native PD-L1 in alveolar LECs isolated from Sp19F-infected MHC-II$^{fl/fl}$ and MHC-II$^{\Delta Epi}$ mice 48hpi. $n \geq 8$ mice, two independent experiments; mean ± SD. **e** Schematic of experimental timeline used. **f** Pie charts for mean frequencies of distinct lung (i.v.CD45.2⁻) CD4⁺ T$_{RM}$ cell- and blood (i.v.CD45.2⁺) CD4⁺ T$_{EM}$ cell- lineages in PD-1$^{+/+}$ and PD-1$^{-/-}$ mice at day 85 as identified using manual gating, Two-way ANOVA with Fisher's LSD test. p-value: #≤ 0.05 for comparison between lung and blood; *≤ 0.05 for genotype-dependent comparison within lungs; $n \geq 5$ mice, two independent experiments. **g** Relative multipotency indices for lung (i.v.CD45.2⁻) CD4⁺ T$_{RM}$ cells (*circles*) and blood such (i.v.CD45.2⁺) CD4⁺ T$_{EM}$ cells (*triangles*) in PD-1$^{+/+}$ (*white*) and PD-1$^{-/-}$ (*red*) mice at day 85, $n \geq 5$ mice, 2 independent experiments, One-way ANOVA with Holm-Sidak's multiple comparison test. All data are presented as mean ± SEM.

performed with compliance to all relevant ethical regulations for animal testing and research, in the United States in accordance with the Guide for the Care and Use of Laboratory Animals published by the National Institutes of Health (after review and approval by the Institutional Animal Care and Use Committee of Boston University) and in Germany according to the German animal protection law (after review and approval by the Regierungspräsidium Darmstadt, Veterinärdezernat of Hesse, Germany).

**Human lung biopsy samples.** De-identified pathologically "disease-free" samples from lung tissue wedge resections or lobectomies were used for experiments. None of the patients included received any checkpoint blockade therapy. The Boston University Medical Center Institutional Review Board reviewed and approved the study, deeming that it did not meet the definition of "human subjects research" (Protocol # H-32271).

**Experimental pneumonia.** Mice were anesthetized via intraperitoneal (i.p.) injection of ketamine and xylazine before infection. To induce experience, *Streptococcus pneumoniae* (*Spn*) suspended in sterile saline was instilled via a 24-gauge angiocatheter directed to the left lobe after surgical exposure of the trachea. Mice were infected intratracheally (i.t.) with $1-5 \times 10^6$ CFU of serotype 19F *Spn* (Strain EF3030) or saline and euthanized 48 h postinfection. To generate *Spn*-specific lung-resident CD4⁺ T$_{RM}$ cells, mice were infected intratracheally (i.t.) with $1-5 \times 10^6$ CFU of serotype 19F *Spn* (Strain EF3030) or saline one week apart, and then allowed to recover for 28–35 days[19]. Mice with such infection histories exhibited a lung-resident CD4⁺ T$_{RM}$ cell-dependent protection on memory recall challenge with a lethal serotype-mismatched *Spn*[19]. To test CD4⁺ T$_{RM}$ cell phenotypes on repeated memory recall, 2X Sp19F experienced mice were infected with $1-3 \times 10^6$ CFU of serotype 23A *Spn* (Sp23A) on Days 35 and 49 before allowing recovery for another 5 weeks. For final challenge, $3-5 \times 10^6$ CFU of serotype Sp35B *Spn* (Sp35B) was intratracheally instilled. Of note, Sp35B (like Sp19F and Sp23A) causes nonlethal pneumonia and was chosen to mimic nonlethal homeostatic bacterial exposures experienced by humans. For analyses of MHC-II and PD-L1 coupling, mice were infected with 50 PFU of Influenza A virus PR8 or $1 \times 10^5$ CFU of *Klebsiella pneumoniae* (ATCC43816).

**Mouse experimental procedures.** For in vivo DQ-Ovalbumin uptake and processing assay, 20 µg of DQ-Ovalbumin, DQ-OVA (Molecular Probes, Cat# D-12053) in 50 µL of sterile saline was instilled i.t. via angiocath directed to the left lung 3 h before euthanasia. For Tamoxifen administration, tamoxifen (Sigma) was dissolved in corn oil (Sigma) to 20 mg/mL stock concentration and stored at 4 °C. Mice were i.p. injected at 100 mg/kg of body weight for five consecutive days. Treated mice were allowed to rest for at least 2 weeks for tamoxifen to wash out before infecting them. For in vivo CD40L blockade, Mice received i.p. injections of 400 µg antimouse CD40L (Clone MR-1, BioXCell, Cat# BE0017-1) or IgG control (BioXCell, Cat# BE0091) antibody on day 6. This was followed by 100 µg i.n. instillations and 250 µg i.p. injections of the respective antibodies on days 7, 9, 11, and 13. For FTY720 administration, Mice were given 1 mg/kg of FTY720 (Sigma) every alternate day by intraperitoneal injections. For recombinant IFN-γ administration, anesthetized mice received a bolus of 100 ng of recombinant IFN-γ (R&D systems) in 100 µL intranasally.

**Murine lung digestion for epithelial cell flow cytometry and FACS sorting.** For high throughput flow cytometry of lung epithelial cells, elastase digestion of murine lung was performed. Briefly, euthanized mice were exsanguinated, trachea cannulated and lungs lavaged thrice with 1 mL Ca-Mg free DPBS containing 0.5 mM EDTA solution. The lungs were then inflated with 1 mL digestion media (RPMI 1640 containing 10% Dextran, 4.5 units/mL of Elastase (Worthington Biochemicals, Lakewood, NJ), and 150 µg/mL of DNase I) and plugged with 0.5 mL of 1% low melting temperature agarose in PBS. The heart-lung blocks were then placed on petri dishes and covered with ice to allow agarose to solidify for at least 5 min. Left lobes of the enzyme-instilled and plugged lungs were dissected and kept for

enzymatic digestion in additional 2 mL of digestion media for 1 h at 37 °C at 200 rpm. The digested lungs were then minced finely using disposable razor blades, washed down with 1 mL digestion media, and transferred to 50 mL conical tubes. The slurry was then kept on a shaker for 20 mins at 37 °C at 200 rpm before adding 10mL of RPMI 1640 with 50% heat-inactivated FBS and 100 µL of 10 mg/mL DNase I. This suspension was then gently vortexed to allow cell aggregate dissociation and incubated on ice for 5 mins. Ten-milliliter RPMI 1640 was then added and the suspension was shaken vigorously at 300 rpm at 37 °C before sequential gravity-mediated filtering through 100 and 70 µm cell strainers. The single-cell suspensions were then centrifuged at 300 g for 10 mins to pellet cells before RBC lysis and resuspension in FACS buffer for flow cytometry as per usual procedures. Cells were blocked with TruStain αCD16/CD32 Fc-Block (BioLegend). Flow cytometry was performed on LSR II Flow Cytometer (BD Biosciences). High-dimensional multiparameter spectral flow cytometry was performed on Aurora (Cytek), SpectraFlo (Cytek) software was utilized for spectral unmixing of the data using an ordinary least square algorithm, and data were analyzed with FlowJo software (BD Biosciences). Gating strategies are provided in the Supplemental figures and were based on use of Fluorescence minus one (FMO) controls.

For data generated using the BASC reporter mice, dispase digestion was used[21]. Briefly, the lungs of euthanized mice were perfused with PBS, cannulated via the trachea, and instilled with 1 ml prewarmed dispase solution (Corning). The trachea was ligated, the lungs extracted from the thorax, and then finely minced using scissors. Two milliliters dispase and DNase I (20 U/ml, Roche) were added to the resulting slurry, and samples digested at 37 °C shaker for 20–30 min. Resulting cell suspensions were gently mixed and sequentially filtered through 100- and 40-µm strainers followed by wash with excess DMEM through the filter. The single-cell suspensions were centrifuged (5 min, 300 g), pellets resuspended in 1 mL pre-cooled FACS buffer. Cell suspensions were resuspended in MACS buffer, mixed with anti-CD45, anti-CD31, and anti-Ter119 microbeads (Miltenyi Biotec), and incubated at 4 °C for 15 min. After washing, cells were loaded onto pre-conditioned MS columns that had been placed in the magnetic field of a MACS separator and the flow-through containing unlabeled cells was collected. These enriched LECs were then blocked using CD16/CD32 Fc-Block (BD Biosciences), stained, and washed before flow cytometry was performed on LSRFortessa Flow Cytometer (BD Biosciences). Gating was based on the use of Fluorescence minus one (FMO) controls. The list of antibodies used in this study are provided in Supplementary Table 7.

For RNA-profiling, epithelial subsets from stained single-cell suspensions isolated from elastase digested lungs were sorted into RPMI 1640 with 20% FBS on ice using FACS-Aria II SORP (BD Biosciences) before proceeding to RNA extraction.

For TEM analyses, stained LEC cell suspensions were fixed using 2% paraformaldehyde in PBS before sorting using the MoFlo Astrios cell sorter (Beckman Coulter).

**Murine lung digestion for leukocyte flow cytometry.** To allow enumeration of extravascular versus the intravascular fraction of lung CD4⁺ T cells, anesthetized mice were retro-orbitally administered 2 µg anti-CD45.2 antibody 3 min prior to euthanasia. Lungs were collected in RPMI 1640 with 10% FBS before processing for flow cytometry. Single-cell suspensions were prepared by digestion of lungs in type 2 collagenase (Worthington Biochemicals, Lakewood, NJ) and DNase I[19]. Cells were blocked with TruStain αCD16/CD32 Fc-Block (BioLegend). Flow cytometry was performed on LSR II Flow Cytometer (BD Biosciences). High-dimensional multiparameter spectral flow cytometry was performed on Aurora (Cytek), SpectraFlo (Cytek) software was utilized for spectral unmixing of the data using ordinary least square algorithm and data were analyzed with FlowJo software (BD Biosciences). Gating strategies are provided in the Supplemental figures and were based on the use of Fluorescence minus one (FMO) controls. The list of antibodies used in this study is provided in Supplementary Table 7.

**Lung processing for human lung flow cytometry.** Pathologically deemed normal segments of lung biopsies obtained from tumors resected from patients were

collected in RPMI 1640. The lung pieces were minced in 2 mL of Digestion media (RPMI containing 10% Dextran, 10 units/mL of elastase, and 150 μg/mL of DNase I) with a disposable razor blade to a fine slurry. The slurry was then washed down with 3 mL more digestion media and transferred to 50 ml conical tubes. The dish was washed with additional 5 mL digestion media and transferred to the 50 mL tube to have a total of 10 mL cell slurry. The suspension was shaken at 250 rpm for 1 h at 37 °C with intermittent vortexing every 20 mins after which 10 mL of RPMI 1640 with 50% heat-inactivated FBS and 200 μL of 10 mg/mL DNAse I was added. The suspension was vortexed to dissociate aggregates and kept on ice for 5 mins. Additional 10 mL RPMI 1640 media was added to bring the final liquid volume to 30 mL and the tube shaken at 300 rpm for 10 min at 37 °C. The lung minces were sequentially filtered through 100 and 70 μm cell strainers using gravity (no scraping). The single-cell suspensions then centrifuged at 300 g for 10 mins to pellet cells before RBC lysis and resuspension in FACS buffer for flow cytometry as per usual procedures. Cells were blocked with Human TruStain FcX- Block (Bio-Legend). Flow cytometry was performed on LSR II Flow Cytometer (BD Biosciences) and data were analyzed with FlowJo software (BD Biosciences). Gating strategies are provided in the Supplemental figures and are based on the use of Fluorescence minus one (FMO) controls. The list of antibodies used in this study is provided in Supplementary Table 7.

**Intracellular cytokine, protein, and transcription factor staining for flow cytometry**. For intracellular staining of transcription factors and Ki67, eBioscience Foxp3/ Transcription Factor Staining Buffer Set (Cat# 00-5523-00) was used as per manufacturer's protocols. For intracellular cytokine staining and subcellular PD-L1 localization, eBioscience Intracellular Fixation & Permeabilization Buffer Set (Cat# 88-8824-00) was used as per manufacturer's protocols. The list of antibodies used in this study is provided in Supplementary Table 7.

**Transmission electron microscopy**. The sorted cells or mouse lungs were fixed for 2 h in 2% glutaraldehyde plus 1% paraformaldehyde in 0.1 M Na Cacodylate buffer, pH 7.4. After washing in buffer, samples were postfixed in 1.5% osmium tetroxide, block stained in 1.5% uranyl acetate, and dehydrated in acetone. They were then embedded in Epon 812, sectioned and stained with uranyl acetate and lead citrate. Images were captured using Tecnai F20 microscope.

**RNA extraction and Real-time PCR**. RNA was extracted from sorted LECs using RNAeasy Micro Kit (Cat# 74004) as per manufacturer's protocols and stored at −80 °C. qRT-PCR was performed using the RNA-to-Ct kit (Life Technologies, Cat# 4392938). Commercially available predesigned TaqMan gene expression assays for *Spc* (Mm00488144_m1), *Scgb1a1* (Mm00442046_m1), *Foxj1* (Mm01267279_m1), *Aqp5* (Mm00437578_m1), *Aw112010* (Mm01197675_m1), *Sox2* (Mm03053810_s1), *H2-K1* (Mm01612247_mH), *Cdkn1a* (Mm00432448_m1), *Cd74* (Mm00658576_m1), *H2-DMa* (Mm00439226_m1), *H2-DMb1* (Mm04213366_s1), *H2-DMb2* (Mm00783 707_s1), *H2-Oa* (Mm00468476_m1), *H2-Ob* (Mm00468801_m1), *Cd274* (Mm030 48248_m1), and 18S rRNA (Cat# 4319413E) from Applied Biosystems were used. The probe context sequence for these assays are detailed in Supplementary Table 8. The quantity of the detectable mRNA was calculated by normalizing to 18S rRNA from the respective sample and then expressed as fold change over mRNA levels of the whole lung of naive mice of pertinent genotype.

**Ex vivo antigen presentation assay**. FACS-sorted LECs from Sp19F-infected mice were co-cultured with OT-II CD4$^+$ T cells (1:10 ratio) with 25μg/mL of OVA$_{323-339}$ peptide (Anaspec) in media containing 25% DMEM (Gibco), 25% F-12 (Gibco), 50% RPMI 1640 (Gibco) supplemented with 10% Heat inactivated FBS, Penicillin/Streptomycin (100 U/mL, Gibco), 1X Non-essential amino acids (Gibco), 2 mM L-Glutamine (Glutamax, Gibco), 1 mM Na-pyruvate (Gibco), 10 mM HEPES (Gibco), 0.2 μM beta-mercaptoethanol, 10 μg/mL Insulin, 50 mM Hydrocortisone, 100 μg/mL Transferrin, 1 mM Na-selenite, and 1 mM beta-estradiol for 8 h at 37 °C 5% CO$_2$ incubator. For C57BL/6J LECs, EasySep™ Mouse CD4$^+$ T Cell negative selection and Isolation Kit (StemCell Technologies, Cat# 19852) was used to enrich OT-II CD4$^+$ T cells from spleens of OT-II TCR transgenic mice and 10 μg/mL of antimouse MHC-II antibody (Biolegend) was used for MHC-II blockade. For assays with MHC-II$^{ΔEpi}$ LECs, EasySep™ Mouse CD4$^+$ T Cell negative selection and Isolation Kit (StemCell Technologies, Cat# 19852) was used to enrich OT-II CD4$^+$ T cells from spleens of OT-II TCR transgenic mice. This was followed by MHC-II blockade of the enriched CD4$^+$ T cell suspension to block any excess MHC-II expressing splenocytes before rigorous washing and coculture with LECs for 8 h.

**Ex vivo stimulation and Intracellular Cytokine staining of CD4$^+$ T cells**. To allow identification of extravascular versus intravascular fraction of lung CD4$^+$ T cells, anesthetized mice were retro-orbitally administered 2 μg anti-CD45 antibody 3 min prior to euthanasia. Single-cell suspension of lung leukocytes was prepared as described before.

For PMA/Ionomycin stimulation, 2 × 10$^6$ cells were stimulated ex vivo in 12 well plate with 250 ng/mL Phorbol Myristate Acetate (PMA)(LC Laboratories, Woburn, MA) and 1.5 μg/mL Ionomycin (Sigma, St. Louis, MO) in T cell stimulation media (RPMI 1640 (Gibco) supplemented with 10% Heat inactivated

FBS, Penicillin/Streptomycin (100U/mL, Gibco), 1X Non-essential amino acids (Gibco), 2 mM L-Glutamine (Glutamax, Gibco), 1 mM Na-pyruvate (Gibco), 10 mM HEPES (Gibco), and 0.2 μM beta-mercaptoethanol) for 1 h at 37 °C and 5% CO$_2$. Monensin (Biolegend, Cat# 420701) and Brefeldin A (Biolegend, Cat# 420601) both at 1X final concentration were added to the cell suspension for last 5 h at 37 °C and 5% CO$_2$. Cells were then processed for intracellular cytokine staining, as per manufacturer's protocols.

For Spn-specific stimulation, 2 × 10$^6$ lung cells were stimulated with beta-propiolactone killed Sp35B (ratio of 1:10 of cells:bacteria) or 1 mg/mL chicken egg ovalbumin (Sigma) ex vivo in 12 well plate for 3 h followed by addition of Monensin and Brefeldin A at 1X final concentration for 14 h at 37 °C and 5% CO$_2$. Cells were then processed for intracellular cytokine staining, as per manufacturer's protocols. The list of antibodies used in this study is provided in Supplementary Table 7.

**Immunofluorescent staining**. The trachea of euthanized mice were cannulated with an 18-gauge angiocath and 1 mL of Tissue-Tek Optimal Cutting Temperature (O.C.T.) compound (Sakura Finetek). Once the lungs were inflated, the left bronchus was tied using a suture, the lung washed in sterile HBSS before embedding in cryomolds with O.C.T. and flash freezing at −80 °C until sectioning. Frozen 8 μm thin coronal sections of the left lungs that contained the whole face of the lungs including the entire airway tree structure were collected for further analyses. Sections with folds and/or tears were rejected. The sections that met our inclusion criteria were first fixed, washed, and permeabilized (with 0.2% Triton X) before blocking with Blocking buffer (PBS with 10% normal donkey serum and 3% BSA) followed by overnight incubation with rabbit antimouse CD4 (abcam, Cat# ab183685) at 4 °C in a humidified chamber. Next day, sections were vigorously washed before incubation with Alexa 594 conjugated Affinipure donkey antirabbit IgG (Jackson Immunoresearch) at room temperature for 1 hr in a dark humidified chamber. All slides were counterstained with DAPI (Molecular Probes by Life Technologies, R37606) before mounting the sections with FluorSave (Millipore Calbiochem: 345789) and covering with coverslip for visualization. Of note, a "no anti-mouse CD4" treated primary control was used to identify true CD4$^+$ events; a section of day 10 mouse lung (i.e., 3 days post 2x Sp19F infection when the lung possesses most CD4 T cells, Fig. 3A) was chosen for the "no anti-CD4 primary control" to ensure that the inclusion criteria for CD4$^+$ events were stringent. Images were captured using a Leica DM4 microscope equipped with Leica DFC 7000T camera and processed using ImageJ 2.0.0-rc-69.

**Quantification of anatomical distribution of CD4$^+$ cells**. Fifteen images of randomly selected fields per coronal section from each mouse lung were identified in a raster fashion, from the top left to bottom right, and were captured at 10x objective with Numerical Aperture of 0.30. Images included autofluorescence of lung tissue in the GFP channel to allow identification of lung anatomical structures. ImageJ was used for automated quantification of lung tissue area identified as airway (along the conducting airways), bronchovascular interstitium (interstitial area around major arteries within the bronchovasculature) and the parenchyma (alveoli) within the GFP channel which was permissible for tissue autofluorescence (AF). Once the areas of distinct lung structures were identified, ImageJ was used to computationally enumerate CD4$^+$ events within the same manually selected, anatomically distinct areas of the imaged fields within the Alexa 594 channel to minimize the inclusion of artefacts as false-positive signals. This was done by setting a standardized minimum threshold value which excluded 99.9% of the background signal on the "no anti-CD4 primary control" slide. Images within Alexa 594 channel underwent watershed segmentation to differentiate overlapping CD4$^+$ events. Once this was done for every one of the 15 images per lung, the number of CD4$^+$ events per unit area of the identified lung structure ($N_X$) was identified as (1):

$$N_X = \frac{\#CD4^+ events_X}{Area_X} \tag{1}$$

where $X$ = the anatomical structure of the lung wherein the CD4$^+$ events were identified.

If at least 1 of the 15 images per mouse lung contained a structure of interest (i.e., an instance of $X$), the dynamics of $N_X$, i.e $D(N_X)$ for that mouse lung was calculated as (2):

$$D(N_X) = \frac{1}{n} \sum_{a=1}^{n} (N_x)_a \tag{2}$$

where $n$ = number of images per lung containing instances of $X$

Since each group contained a total of at least four mice per time points collected across two experiments, the group dynamics of $N_X$ for the whole group of mice at their specific time points, i.e., $G(D(N_X))$ was then calculated as (3):

$$G(D(N_X)) = \frac{1}{m} \sum_{b=1}^{m} [D(N_X)]_b \tag{3}$$

where $m$ = total number of mice per group for that timepoint.

**Algorithmic analysis of single-cell fluorescence datasets**. Data processing pipeline was established using the Omiq.ai cloud computation platform (Omiq). For temporal LEC MSFC data analysis, live CD45$^-$EpCAM$^+$ single-cell datasets

containing equal quantities (25,000 events per animal) of cells from all experimental timepoints were concatenated and asinh transformed (cofactor = 6000). Datapoints were projected into two-dimensional space with opt-SNE algorithm[26] incorporating fluorescence, forward scatter and side scatter parameters (perplexity = 50, theta = 0.5, opt-SNE endpoint = 7500; PCA preinitialization embedding). Intensities of each fluorescence parameter were overlaid to represent expression levels. Same dataset was clustered with Phenograph algorithm ($k = 20$, distance metric = euclidean)[27]. Groupings of clusters based on hierarchical clustering of median fluorescence intensities (MFI) across multiple protein markers were annotated and color-overlaid on the opt-SNE projection of multidimensional data. For T cell flow cytometry profiling, live CD45$^+$CD4$^+$CD19$^-$CD45.2$^+$ for lung intravascular ('blood') cells or live CD45$^+$CD4$^+$CD19$^-$CD45.2$^-$ for lung extravascular ('lung') T cells were concatenated, asinh transformed (cofactor = 6000) and clustered with Phenograph ($k = 20$, distance metric = euclidean). CD44$^+$CD11a$^+$ clusters (as determined based on the MFI cutoff) were subsampled as memory T cell data. These data were re-clustered with Phenograph ($k = 20$, distance metric = euclidean) and projected into opt-SNE space (perplexity = 30, theta = 0.5, opt-SNE endpoint = 5000; PCA preinitialization embedding). Clusters were color-coded and overlaid on the opt-SNE projections. Each marker MFIs of Phenograph clustered datasets were organized into hierarchically clustered heatmaps. Clusters were classified as positive for specific lineage-defining transcription factors (LDTF) based on the MFI value cutoffs set by measuring the MFIs of LDTF-negative non-T cell populations sampled from the same dataset. This approach was preferred over the FMO-based cutoff calculation to alleviate the MFI difference caused by nonspecific binding of anti-LDTF antibodies to cells that do not express them. Frequencies of each cell type and MFI values for individual markers were calculated from corresponding clusters per each animal and data were plotted and compared using Prism 8.0 (Graphpad).

**Quantification of Relative Multipotency Index.** Relative multipotency indices for CD4$^+$ CD44$^+$CD11a$^+$ memory T cells in different anatomical sites were calculated as (4):

$$Relative\ Multipotency\ Index = ln\left(\frac{\%\ of\ memory\ CD4 + T\ cells\ expressing \geq 2\ LDTFs}{\%\ of\ memory\ CD4 + T\ cells\ expressing\ 1\ LDTF}\right) \quad (4)$$

Where memory CD4+ T cells = CD4$^+$CD62L$^-$CD44$^+$CD11a$^+$ CD4$^+$ T cells, and LDTF = lineage-defining transcription factors.

i.v.CD45.2$^-$ CD4$^+$ memory T cells were identified as the lung memory T cells while the i.v.CD45.2$^+$ CD4$^+$ memory T cells were identified as the blood memory T cells.

**Statistical Analyses.** Statistical analyses were performed using Prism 8.0 (GraphPad). Differences were deemed statistically significant if the $p$ value or FDR $q$ value was ≤0.05. Each figure legend communicates the number of mice used per experiment, the number of experiment replicates performed and the statistical tests used to make comparisons. For all figures, data are represented as mean ± SEM.

**Reporting Summary.** Further information on research design is available in the Nature Research Reporting Summary linked to this article.

## Data availability

Original data generated in this study are available within the paper and associated supplementary files, with additional information available from the corresponding author upon reasonable request. Source data are provided with this paper.

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

## Acknowledgements

We thank Riley Pihl and the Boston University School of Medicine Flow Cytometry Core Facility (BU-FCCF) for assistance with flow cytometry and FACS sorting. We thank Dr. Esther Bullitt for assistance with electron microscopy and Dr. Mary Williams for her expertize with electron microscopy of LECs. We would also like to thank Dr. Jason Rock, Dr. Anna Engler and Hanne Richardson for advice with tamoxifen procedures and Cheryl Spencer for help with acquisition of consented human lung samples. Finally, we would like to thank all the consenting donors for their generosity. This work was supported by NIH grants including HL147397 to EIA, HL142199 to KAB, HL147461 to FTK, HL136725 to MRJ, GM120060 and HL111449 to LJQ, AI115053, HL135756, and HL137081 to JPM and T32 HL007035 for support of trainees in addition to support by the German Research Foundation (DFG) Clinical Research Group KFO309 TP08, the Excellence Cluster Cardio-Pulmonary Institute (CPI), and the German Center for Lung Research (DLZ) to TB.

## Author contributions

A.T.S. and J.P.M. conceived the project, designed the experiments and supervised the study. A.T.S. performed experiments and analyzed the data with assistance from C.L.D.A., E.I.A., K.A.B., F.T.K., A.R., N.S.E., A.M.S., I.M.C.M., B.R.T., A.H., W.N.G. Experiments with BASC-reporter mice were performed by I.S. with guidance from T.B. A.C.B. provided guidance with multispectral cytometry and performed computational analyses including the Phenograph clustering and optSNE visualization on the data. H.K., T.B., M.R.J., L.J.Q., A.C.B. and J.P.M. contributed resources used in this study. A.T.S. and J.P.M. wrote the manuscript, which was edited and approved by C.L.D.A., E.I.A., K.A.B., I.S., F.T.K., A.R., N.S.E., A.M.S., I.M.C.M., B.R.T., A.H., W.N.G., H.K., T.B., M.R.J., L.J.Q. and A.C.B.

## Competing interests

The authors declare no competing interests.
