## [Peer Review File · Nature Communications]

Antigen presentation by lung epithelial cells directs CD4+ TRM cell function and regulates barrier immunityREVIEWER COMMENTS

Reviewer #1 (Remarks to the Author):

This manuscript has some very detailed phenotyping data on immune cells, in particular tissue-resident memory T (TRM) cells, in the lung of mice either lacking MHC class II on epithelial cells or after infection with *Streptococcus pneumoniae* (Spn) and concludes that antigen presentation by epithelial cells is "a critical regulator of CD4+ TRM cell activities".

While the data looks very spectacular, the study tells is very little about the role of TRM cells in the control of Spn infection. There is not a single experiment that shows that epithelial cells act as APC for T cells and that this presentation is critical for activation of lung TRM cells and that these lung TRM cells actually confer protection against Spn infection.

The phenotyping data is so detailed that it makes it almost impossible to see the message from each figure and indeed from the whole paper. Sometime less is more and in this case a lot of the data could go in supplemental figures and the key points in the main figures. For example, we are asked to compare data on Fig 5C with what was presented in Fig 4H to compare resting with infected lungs; this would be so much better if presented on the same figure.

In conclusion, the data on phenotyping needs to be condensed and data on cell function, antigen presentation and role in protective immunity needs to be considerably expanded.

Reviewer #2 (Remarks to the Author):

Shenoy et al identify MHC II on LEC (including a novel MHCII^{high} LEC population) as critical for the regulation of CD4 TRM cells during recurrent exposure to pathogens. The authors propose MHCII⁺ LE cells act as APC for CD4 TRM cells, although limited characterization of APC function is presented. They show that co-stimulatory/co-inhibitory molecules are differently expressed by LEC subtypes, with MHCII^{high} LEC co-expressing PD-L1 under the conditions studied. Using murine lung infection models, multi-spectral flow cytometry, and immunofluorescence, the authors nicely track the development/location of the TRM niche at conducting airways and show a temporal relationship with expression of PD-L1 by LEC during infection. They show that the MHCII expression levels in LECs modulate TRM cell plasticity, although the mechanism is not deciphered. The authors further used a mismatched lung infection model to investigate the effects of LEC MHCII on the TRM recall response and its apparent regulation of lung infiltration by peripheral immune cells. Interestingly in this regard, in animals with LE-specific KO of class II and a consequent lack of PD-L1 on the cell surface, they find a phenotype that resembles that seen in patients on checkpoint blockade. Overall, the experiments in this predominantly descriptive study are well-designed and approaches and results are clearly explained in the text. The data strongly support that loss of MHCII on LEC has effects on CD4 TRM. I have some suggestions for the authors.

1. In the interpretation of Figure 1 data, the authors argue that the presence of lamellar bodies indicates the alveolar localization of the MHChigh LEC. Can the authors do a histological analysis to directly localize these interesting cells? This is relevant to the referenced to anatomic separation of LEC subsets.
2. Data in Figure 2A histogram plots indicate higher MFI for DQ-OVA for several LEC types after infection compared to the naïve counterparts; however, the bar-graph quantification indicates otherwise. If any normalization has been performed for the evaluation, it has not been mentioned. Please explain the difference.
3. The data in Figure 2 to support "antigen presenting abilities" are quite limited. For example, the results are insufficient to determine whether these cells express various accessory molecules that regulate class II mediated antigen presentation. Further characterization of these cells as APC would strengthen the paper.
4. The authors suggest a stable coupling of constitutive MHCII with PD-L1 expression on AT2 and MHChigh cells. Have they looked at other conditions besides Spn infection?

5. The statement that the transient increase in CD4 T cells (Fig 3A) represents non-specific T cells is not directly supported, though it is probably true. Soften the statement?
6. In Fig 3I, the differences in levels of CD4 T cells in the bronchovascular interstitium and airway when MHCII is not expressed in LEC (Nkx2.1 expressing cells) though significant, are quite modest. Can the authors support their conclusion with additional (orthogonal) data?
7. The effect of blocking CD40/CD40L is more robust. However, the mechanism ("antigen presentation by LEC) is not clear given that CD40 is expressed on other cells besides club and multiciliated LEC (e.g., DC).
8. To better understand the effect of LEC MHCII on TRM:
 - a. It would be beneficial to see the effects of MHCII Δ Epi on Teff, TCM, and TEM cells. Do LEC MHCII cells establish and maintain the plasticity of lineage-specified TRM cells or does LEC MHCII expression affect the development of TRM cells through the intermediate stages.
 - b. The effect of class II loss on TRM plasticity is interesting. However, the mechanism (and whether for example the cells now interact with different lung class II+ APC) is not deciphered. A more mechanistic understanding would strengthen the paper.
9. In Figures 5 and 6, the effects of MHCII deficiency in LEC on TRM cells have been evaluated by cytokine production and innate immune instruction. However, a key feature of Sp infection is also iBALT formation, which can suggest the pathogen-specificity of the CD4+ TRM cells. In evaluating the effects of LEC MHCII on CD4+ TRM maintenance and recall function, pathogen-specificity is an important factor. An experiment to address this specificity question will strengthen the study.
10. Figure 6C: The changes in distribution of cells are shown. Do absolute numbers of the different cell types change to a similar degree?
11. Based on Figure 7, the authors showed a strong correlation between MHCII and PD-L1 expression in human lungs and inhibition of surface presentation of PD-L1 in LEC lacking MHCII in the murine model. However, direct evidence linking the absence of MHCII to the expression and localization of PD-L1 is lacking, leaving it unclear whether the differential expression and localization of PD-L1 is a direct or indirect effect of MHCII expression in LECs. A more elaborate experiment exploring the molecular factors would add significantly to this observation.

Minor Comments

1. The MHChigh LEC exhibit some features of lung progenitor cells. More comprehensive characterization of the "stem-ness" of these cells will be of interest. Here, can the authors comment on how they think these cells compare to the AT2-signalling stem cells reported in Travaglini et al, Nature, 2020.
2. In Figure S1B: the gating strategy for DC would seem to identify a subset of DCs (there are CD24neg cDC2 in the lung) and potentially include B cells. While this may not be critical for a comparison of class II levels with LEC, have the authors used other markers to determine the identity of the gated cells?
3. Figure 2F-K: statistical annotation is missing from plots. Also, please convert the y-axes of Figure 2F and 2I to decimal to be consistent with the other data in the same panel.
4. Figure S4: Legend for panel C is missing, so C-E letters in legend are also incorrect
5. Figure S5B: is the cell/label correct as MHChi Club?
6. Fig 3D: the results for Teff are missing. This effects the statement that only TRM are still PD-1+ at day 35.
7. Figure 3I: Is the IF image from day 56, but the quantitation from day 35? Legend also says one-way ANOVA, but the comparisons are each between only 2 groups.
8. Line 123 Should be ..., which are...
9. Line 181 The modulation of APC molecules on LECs on day 8 was microbe-induced,
10. Line 196 snapshots of the dynamics
11. Line 705 needs editing

Reviewer #3 (Remarks to the Author):

The manuscript "Epithelial MHC-II governs CD4+ TRM cells and barrier immunity" examines the impact of MHC-II expression on lung CD4+ TRM location, plasticity and the lung immune landscape. Using SPC-GFP mice, electron microscopy, and MFSC analysis, the authors

characterized lung epithelial cells, including a population of MHC-II^{high} LEC with a novel phenotype distinct from both club cells and traditional AT2 cells. They demonstrated that all LEC exhibit antigen presenting ability, but that following Spn infection, costimulatory and coinhibitory molecule expression by LEC is dynamic and cell-type specific. The authors suggest that lung epithelial MHC-II facilitates CD4⁺ TRM formation, since MHC-II Δ epi mice have smaller CD4⁺ TRM niches along the airway epithelium and within bronchovascular interstitium. Additionally, LEC-MHCII also regulates the phenotype of CD4⁺ TRM. Again, using MSFC analysis, the authors found that following repeated exposure to a serotype-mismatched strain of Spn, CD4⁺ TRM are heterogeneous and express multiple lineage defining transcription factors. Additionally, CD4⁺ TRM expressing multiple LDTFs increase in the absence of LEC MHCII. In contrast, circulating memory CD4⁺ T cells do not exhibit as much plasticity. Following re-stimulation, T cell clusters express fewer LDTFs, and in the absence of LEC MHC-II, lung CD4⁺ T cells have increased Th17 response, but decreased Th1 response correlating with dysregulation in immune barrier landscape. Finally, the authors noted similarities in the immune dysregulation of infected MHC-II Δ epi mice and checkpoint blockade immunotherapy patients. They demonstrated that LEC MHC-II expression correlates with PD-L1 expression across all LEC cell types in mice as well as by LEC in humans. Additionally, LEC from MHC-II Δ epi lack PD-L1 expression, suggesting a lockstep between MHC-II and PD-L1 on LECs. In the absence of MHC-II, PD-L1 was trapped in the intracellular compartment and degraded.

The authors present novel data examining the effect of lung epithelial cell MHC-II expression on CD4⁺ TRM localization, plasticity and immune response following challenge that may be of importance to the general community. The is a well-executed study. However, a few comments and suggestions might help to support the conclusions and clarify the significance of these findings:

1. The authors use H2-Ab^{f1}/f1 mice crossed to Nkx2.1Cre-ERT2 mice to examine the requirement of MHCII expression by LEC. The authors demonstrate that MHC-II remains intact on CD45⁺ leukocytes following administration of tamoxifen. Is MHCII expressed by other CD45⁻ cells at baseline or following activation in these mice following administration of tamoxifen?
2. The section title "a novel population of LECs" might be a bit strong. Might these cells be a subset?
3. Does the plasticity of CD4⁺ TRM depend on the Spc serotype used for infection? For example, in Figure 4, if memory recalls were performed with Sp35B rather than with Sp23A, but analyses were still performed at day 105, would the CD4⁺ TRM still exhibit the same plasticity? Also, what is LDTF expression by TRM on day 56 before memory recalls?
4. Can memory recall induce formation of non-antigen-specific lung CD4⁺ TRM? In Figure 5H, if cells are stimulated with antigen-pulsed APCs rather than PMA and ionomycin, is the same pattern of cytokine secretion observed?

Minor comments:

1. In Figure 1B, it might be nice to have a legend for the flow plot indicating that the dark blue population is dendritic cells.
2. In Figure 3A, was FoxP3 analyzed? Could the CD25⁺ cells be Treg?
3. Is it possible that contraction of CD4⁺ T cells in the parenchyma results from lack of costimulatory receptors by LEC?
4. It might be helpful for readers without a lung biology background to have a supplementary figure with a schematic mapping the locations of LEC cell types (perhaps with an indication of their MHC and costimulatory expression) and CD4⁺ T cells over time following infection.

Reviewer #1 (Remarks to the Author):

This manuscript has some very detailed phenotyping data on immune cells, in particular tissue-resident memory T (TRM) cells, in the lung of mice either lacking MHC class II on epithelial cells or after infection with *Streptococcus pneumoniae* (Spn) and concludes that antigen presentation by epithelial cells is “a critical regulator of CD4⁺ TRM cell activities”.

While the data looks very spectacular, the study tells is very little about the role of TRM cells in the control of Spn infection. There is not a single experiment that shows that epithelial cells act as APC for T cells and that this presentation is critical for activation of lung TRM cells and that these lung TRM cells actually confer protection against Spn infection.

Although not the focus of this current manuscript, there is much known about specific roles of T_{RM} cells in the control of *Spn* infection. We have previously demonstrated that Sp19F experience (like in the present manuscript) seeds the lungs with T_H17-polarized CD4⁺ T_{RM} cells, which are essential to the multi-log improvement in lung defense against subsequent *Spn* infections of a different serotype (Smith *et al*, *Mucosal Immunology* 2018). The CD4⁺ T_{RM} cells confer this protection by accelerating neutrophil recruitment in the early hours after heterotypic infection, mediated by increased epithelial production of CXCL5, in response to CD4⁺ T_{RM} cell-derived IL-17A that helps stabilize mRNA transcripts including CXCL5 in the lung epithelial cells (Shenoy *et al*, *Mucosal Immunology* 2020). With CD4⁺ T_{RM} cells established as protective agents of immunity against heterotypic *Spn* infections, via known mechanisms, our current study aimed to discover upstream factors that instruct CD4⁺ T_{RM} cells in the lung. In the present manuscript, we define novel roles for lung epithelial cells in instructing lung CD4⁺ T_{RM} cells, with distinct epithelial subsets using MHC-II and cell-specific co-stimulatory/co-inhibitory molecules to localize and skew the T_{RM} cells in the lung. The new **Supplementary Figure 19** schematizes some of these findings. Although the accelerated neutrophil recruitment and improved lung defense can occur in the absence of this CD4⁺ T cell instruction (**Figures 5a-b** in the revision), it contributes to other aspects of the immune responses in lungs with *Spn* infection, enhancing monocyte recruitment and differentiation, constraining eosinophilia, and skewing the immune landscape away from Th17 and towards Th1 phenotype (**Figures 5c-d** in the revision). This lung CD4⁺ T_{RM} cell instruction has profound effects on non-infectious chronic pulmonary diseases as well, based on ongoing studies of ours that are outside the scope of the present manuscript. The fundamental lung immunology discoveries reported here are the beginning for establishing the functional significance of lung epithelial instruction of CD4⁺ T_{RM} cells, which applies to lung immunity against infection as well as to chronic pulmonary diseases, which all depend on lung immunity (as broadly discussed in Mizgerd, *AJRCCM* 2012). Roles of T_{RM} in the control of *Spn* infection are previously reported (Smith *et al*, *Mucosal Immunology* 2018; Shenoy *et al*, *Mucosal Immunology* 2020). Roles of lung epithelial cells in directing CD4⁺ T_{RM} cells and lung immunity are reported in the present manuscript, including measures

of integrated lung defense and elucidation of pulmonary responses that depend on this epithelial cell influences on CD4⁺ T cell biology (**Figure 5** and **Supplementary Figures 6, 13, 14, and 15** in the revision).

In response to these reviewer comments about antigen presentation, we extensively revised the manuscript to include new studies which now directly show that epithelial cells act as APC for T cells. These new experiments used an *ex vivo* co-culture system to demonstrate that FACS-sorted LECs are capable of presenting OVA₃₂₃₋₃₃₉ peptide to activate OT-II TCR-transgenic CD4⁺ T cells, which is abrogated by MHC-II blockade hence MHC-II-dependent (**Supplementary Figure 2b** and its legend plus **pages 7 and 31-32** in the revision). Expanding upon this, additional new experiments show that FACS-sorted LECs isolated from MHC-II^{fl/fl} mice are adept at activating OT-II TCR CD4⁺ T cells by presenting OVA₃₂₃₋₃₃₉ peptide, but the genetic ablation of MHC-II from those LECs in MHC-II^{ΔEpi} mice eliminates this ability (**Figure 2c** plus **pages 7, 31-32, and 45-46** in the revision). These revisions to the manuscript substantially improve the manuscript, together unequivocally showing that LECs can act as APCs for CD4⁺ T cells, with epithelial MHC-II critical for T cell activation.

The re-activation of lung T_{RM} cells may not require epithelial MHC-II. We measured pulmonary bacterial burden, IL-17A, and IFN-γ levels as reports of memory recall T cell activation (**Supplementary Figures 6a-d** in the revision). At this early 8-hour time-point, there is minimal contribution of circulating T cells and both the protective effects mediated by CD4⁺ T_{RM} cells and the pulmonary IL-17A and IFN-γ depend on CD4⁺ T cells (Smith *et al*, *Mucosal Immunology* 2018; Shenoy *et al*, *Mucosal Immunology* 2020). We did not observe genotype dependent differences in bacterial burdens, IL-17A, or IFN-γ levels, suggesting that antigen presentation by LECs is not critical for this recall activation step in the lungs (**Supplementary Figures 6a-d** in the revision). Other APCs in the lung, such as macrophages, dendritic cells, B_{RM} cells, and more, may do so instead. Rather than activating the lung T_{RM} cells during recall infections, MHC-II on LECs is required for dictating the numbers and locations of lung T_{RM} cells after recovery from infection (**Figure 3** in the revision), their transcription factor phenotypes and plasticity in the lung (**Figure 4** and **Supplementary Figures 7-8, 10, and 12** in the revision), and the integrated pulmonary immune response during recall infections (**Figure 5** and **Supplementary Figures 13 and 14** in the revision).

The phenotyping data is so detailed that it makes it almost impossible to see the message from each figure and indeed from the whole paper. Sometime less is more and in this case a lot of the data could go in supplemental figures and the key points in the main figures. For example, we are asked to compare data on Fig 5C with what was presented in Fig 4H to compare resting with infected lungs; this would be so much better if presented on the same figure.

Revised as suggested. We definitely appreciate the reviewer's advice, agree, and have worked to address the concern. We substantially streamlined the presentation

of T_{RM} phenotyping data. The original Figure 4 and Figure 5 were merged in the revision, allowing key data from the two different but related sets of experiments to be viewed in context of each other within a single figure (**Figure 4** in the revision). Multiple figure panels from the original versions of these T_{RM} phenotyping data figures were transferred to supplementary figures in the revision (all of the original Figures 4b, 4e, 4h, 5b, 5c, 5d, 5e, 5g are now supplemental figures). In addition, we endeavored to follow this advice elsewhere, beyond T_{RM} phenotyping, and also transferred the original Figures 1e, 3h, 7e, and 7g to becoming Supplemental Figures in the revision. This streamlining should help emphasize the message and key findings. Related to this thrust to better communicate the paper's message, we hope that the new **Supplementary Figure 19**, schematizing some take-away messages, is a further helpful step towards this goal.

In conclusion, the data on phenotyping needs to be condensed and data on cell function, antigen presentation and role in protective immunity needs to be considerably expanded.

Revised as suggested, as detailed in the prior two point-by-point responses. The data on phenotyping was condensed into a single Figure 4. The requested expansions from Reviewer 1 were added, along with expansions to address related comments from other reviewers, resulting in a total of more than 33 new figure panels (in **Figures 1g-h, 2c, 6e-g, SF1c, SF2b, SF2d-e, SF6a-f, SF7a-b, SF8a-e, SF9a-g, SF12, SF14, SF15b, SF17a-b, SF18, and SF19a-c**) that more expansively flesh out APC functions for LEC MHC-II and their significance and mechanisms of action as directors of CD4⁺ T cells in the lung.

Reviewer #2 (Remarks to the Author):

Shenoy et al identify MHC II on LEC (including a novel MHCII^{high} LEC population) as critical for the regulation of CD4 TRM cells during recurrent exposure to pathogens. The authors propose MHCII⁺ LEC cells act as APC for CD4 TRM cells, although limited characterization of APC function is presented. They show that co-stimulatory/co-inhibitory molecules are differently expressed by LEC subtypes, with MHCII^{high} LEC co-expressing PD-L1 under the conditions studied. Using murine lung infection models, multi-spectral flow cytometry, and immunofluorescence, the authors nicely track the development/location of the TRM niche at conducting airways and show a temporal relationship with expression of PD-L1 by LEC during infection. They show that the MHCII expression levels in LECs modulate TRM cell plasticity, although the mechanism is not deciphered. The authors further used a mismatched lung infection model to investigate the effects of LEC MHCII on the TRM recall response and its apparent regulation of lung infiltration by peripheral immune cells. Interestingly in this regard, in animals with LEC-specific KO of class II and a consequent lack of PD-L1 on the cell surface, they find a phenotype that resembles that seen in patients on checkpoint blockade. Overall, the experiments in this predominantly descriptive study are well-designed and approaches and results are clearly explained in the text. The data strongly support that loss of MHCII on LEC has effects on CD4 TRM. I have some suggestions for the authors.

1. In the interpretation of Figure 1 data, the authors argue that the presence of lamellar bodies indicates the alveolar localization of the MHCII^{high} LEC. Can the authors do a histological analysis to directly localize these interesting cells? This is relevant to the referenced to anatomic separation of LEC subsets.

We share the reviewer's enthusiasm for learning more about these SPC^{low}MHC^{high} LECs, including their location in the lower respiratory tract. In response, we performed new experiments to revise and improve the manuscript. Complementing the TEM analyses of sorted cells in the original manuscript, we performed TEM analyses of intact mouse lungs for the revision, as suggested. Lamellar bodies were readily identifiable. Imaging of >100 anatomically distinct areas of lung sections from 6 different C57BL/6 mice revealed consistent localization of lamellar bodies to cells within the alveolar septae. No cells containing lamellar bodies were observed in the conducting airways. Lamellar body-containing cells observed in the alveoli often contained electron dense organelles consistent with club cell secretory granules. Such structures were abundant in the club cells of the TEM from mouse lungs. However, the morphology of these organelles was not sufficiently distinctive that lamellar body-containing cells in alveoli could be segregated into positive and negative for these structures. These data support more firmly our original conclusion that the SPC^{low}MHC^{high} LECs are in the alveoli specifically. Results from these TEM analyses were communicated in the new **Supplemental Figure 1c** and its legend of the revision and presented in the text on **pages 6 and 31**. Bronchioalveolar stem cells (BASCs) share some features

(lamellar bodies, low levels of SPC, low levels of Scgb1a1 transcripts) with our SPC^{low}MHC^{high} LECs, and have been localized in the lung (to the edges of alveoli that connect with bronchioles). Thus, we tested whether our cells might overlap with BASCs and therefore localize to these junctions. To address this question, we created a new collaboration. Along with Dr. Braun in Germany, we leveraged genetically engineered mice that express YFP and mCherry from the mouse loci for SPC and Scgb1A1, respectively (Salwig *et al*, *EMBO Journal* 2019). These mice provide an independent means of assessing SPC expression in mouse LECs, and allowed direct comparisons of SPC^{low}MHC^{high} LECs with BASC cells. Flow cytometry on dispase-digested lungs from the transgenic mice independently confirmed a distinct LEC subset that expressed low levels of SPC and high levels of MHC-II (**Figure 1g** in the revision). The SPC^{low}MHC^{high} LECs were distinct from club cells and from AT2 cells, as in the initial version of the manuscript. The transgenic mice also clearly distinguished these cells from BASCs, which was new information and an important addition to the manuscript. Further, we discovered that BASCs express MHC-II at intermediate levels that fall between the high expression in alveoli (on AT2 cells and SPC^{low}MHC^{high} LECs) and the low expression in airways (on club cells and multiciliated cells), reinforcing our proposed model of a structured MHC-II gradient along the epithelial surface of the respiratory tract, increasing towards the periphery. Using these transgenic reporter mice, the SPC-YFP signal was observed to be restricted to alveoli, like lamellar bodies in the TEM analyses, again supporting an alveolar localization for the SPC^{low}MHC^{high} LECs and revealing that the SPC^{low}MHC^{high} LECs must therefore be distal to the junctional BASCs (**Figure 1h** in the revision). Revisions discussing the new studies with the transgenic reporter mice were included as **Figures 1g-h** and text on **pages 5-6, 26, 28, and 45**.

2. Data in Figure 2A histogram plots indicate higher MFI for DQ-OVA for several LEC types after infection compared to the naïve counterparts; however, the bar-graph quantification indicates otherwise. If any normalization has been performed for the evaluation, it has not been mentioned. Please explain the difference.

We apologize for any confusion. The quantitative data in the summary graphs, on the right side of Figure 2a, show data from multiple mice and multiple independent experiments and communicate MFI for each individual mouse with no further data normalization. The histograms were intended to show representative data from a single mouse within each group, but there is (as shown in the bar graphs) variation within groups and no two mice look identical or are perfect representatives. Guessing that the histograms from the club cells and the AT1 cells might have been the specific data that gave the misleading impression that MFIs from infected mice might have been higher than in the uninfected mice, we revised these panels to show different examples for those cells in **Figure 2a** of the revision. As in the original version, the summary graphs convey all the data including non-normalized MFI values and the statistical analyses of comparisons between infected and uninfected groups (**Figure 2a** in the revision).

3. The data in Figure 2 to support “antigen presenting abilities” are quite limited. For example, the results are insufficient to determine whether these cells express various accessory molecules that regulate class II mediated antigen presentation. Further characterization of these cells as APC would strengthen the paper.

We revised as suggested to determine whether these cells express various accessory molecules that regulate class II-mediated antigen presentation and to further characterize these cells as APCs. We performed three new major experiments to address these concerns. To examine relevant accessory molecules, we performed qRT-PCR analyses for the expression of CD74, H2-DMA, H2-DMb1, H2-DMb2, H2-Oa, and H2-Ob in FACS-sorted LECs from SPC-GFP mice primed for antigen expression by 48 hours of Sp19F infection. We included lung CD11c⁺ cells and lung PDGFR α ⁺ cells as positive and negative controls, respectively. Results revealed that all LECs expressed accessory molecules at higher levels than the negative control PDGFR α ⁺ cells and often as high or higher than the positive control CD11c⁺ cells, with previously unknown variation among LECs in these expression levels (**Supplementary Figure 2a** as well as **page 7** and **31** in the revision). LECs consistently expressed the CD74 invariant chain and H2-DM positive regulators of MHC-II antigen loading, while the H2-DO negative regulators were less prominent except for in multiciliated cells. These new data reveal that LECs are appropriately armed for accessory function. We directly tested for antigen presentation by all 4 LEC populations using *ex vivo* cultures of FACS-sorted LECs with OVA₃₂₃₋₃₃₉ peptide and CD4⁺ T cells transgenic for TCR specific to this OVA peptide. LECs purified from normal C57BL/6 mice were capable of activating OT-II TCR transgenic CD4⁺ T cells, which was completely inhibited by MHC-II blockade (**Supplementary Figure 2b** as well as **pages 7** and **31-32** in the revision). Even more definitively, the OVA peptide-driven activation of OT-II CD4⁺ T cells cultured with sorted LECs was also completely abrogated by deletion of MHC-II specifically targeted to the lung epithelium using the Nkx2.1-CreERT2 system (**Figure 2c** as well as **pages 7, 31-32, and 45-46** in the revision). These new studies substantially improve the manuscript by directly testing and confirming that LECs use the MHC-II system to present antigen and thereby activate CD4⁺ T cells, making them *bona fide* APCs.

4. The authors suggest a stable coupling of constitutive MHCII with PD-L1 expression on AT2 and MHChigh cells. Have they looked at other conditions besides Spn infection?

We thank the reviewer for this intriguing question. In response, we performed new experiments to test applicability of the lockstep between surface MHC-II and PD-L1 expression in other conditions, with lungs challenged by influenza A virus, the gram-negative bacteria *Klebsiella pneumoniae*, or recombinant murine IFN- γ (**Supplementary Figure 17a** and its legend as well as **pages 19, 27, 32** in the revision). Different stimuli had different effects on different LECs, but surface MHC-II and PD-L1 on LEC subsets consistently displayed the association we

originally discovered using the gram-positive bacteria *Spn* (**Supplementary Figure 17a** and its legend as well as **pages 19, 27, 32** in the revision). To test for dependency, we challenged tamoxifen treated MHC-II^{fl/fl} and MHC-II^{ΔEpi} mice with recombinant IFN-γ to determine whether the high LEC PD-L1 levels depended on LEC MHC-II (**Supplementary Figure 17b** and its legend as well as **pages 19, 27, 32** in the revision). Consistent with our initial conclusion, PD-L1 was absent on the surfaces of alveolar LECs from MHC-II^{ΔEpi} compared to MHC-II^{fl/fl} mice, even under pressure from recombinant murine IFN-γ. These findings bolster our postulate that alveolar LEC MHC-II and PD-L1 are in a lockstep, and demonstrate that this new concept applies more broadly than to *Spn* alone.

5. The statement that the transient increase in CD4 T cells (Fig 3A) represents non-specific T cells is not directly supported, though it is probably true. Soften the statement?

Revised as suggested to communicate that the lack of antigen specificity for these early T cells was inferred but not known (**pages 9-10** in the revision).

6. In Fig 3i, the differences in levels of CD4 T cells in the bronchovascular interstitium and airway when MHCII is not expressed in LEC (Nkx2.1 expressing cells) though significant, are quite modest. Can the authors support their conclusion with additional (orthogonal) data?

We agree that differences in the original Figure 3i (Figure 3h in the revision) are modest in scope. The approach used, quantitative immunofluorescence within histologic sections, is the gold standard for quantifying T_{RM} cells in tissue niches, established by David Masopust and colleagues (Steinert *et al*, *Cell* 2015). Immunofluorescence followed by mathematical scaling is the most reliable, accurate, and sensitive method to enumerate location-specific T_{RM} cell numbers. Relating to rigor (including both maximizing reproducibility and minimizing bias), the data in Figure 3h in the revision were products of 2 completely independent experiments from start to finish, each with 3-4 mice per group that together yielded 6-8 mice per group which were each treated as individual datapoints (each mouse becoming a single N), with each individual datapoint from a given mouse being generated from data collected over 15 separate randomly captured images from the sectioned lungs of that mouse, with the fluorescent signals in each image from each section quantified using computer software rather than a human observer, and with the human guiding that software blinded to the identity of the mouse lung being analyzed. Finally, we wish to emphasize that the T_{RM} cell numbers in Fig 3h are one of many assessments of T_{RM} cell biology throughout the manuscript, and multiple orthogonal lines of evidence were presented to support the conclusion that lung T_{RM} cell biology depends on the presence of lung epithelial cell MHC-II (e.g., Figures 4a-f plus Supplementary Figures 7a, 7c, 8a-d, 10a-d, and 12 in the revision).

7. The effect of blocking CD40/CD40L is more robust. However, the mechanism

(“antigen presentation by LEC) is not clear given that CD40 is expressed on other cells besides club and multiciliated LEC (e.g., DC).

We agree and revised the text to clarify that CD40 is also expressed on other cells besides club and multiciliated LEC (page 11).

8. To better understand the effect of LEC MHCII on TRM:

a. It would be beneficial to see the effects of on Teff, TCM, and TEM cells. Do LEC MHCII cells establish and maintain the plasticity of lineage-specified TRM cells or does LEC MHCII expression affect the development of TRM cells through the intermediate stages.

We have revised the manuscript to better consider effects of LEC MHC-II on T cell subsets and their inter-relations, as suggested. By adding CD62L into our Phenograph clustering and heat map datasets, the revised presentation allows separate analyses of T_{CM} cells (CD44+CD62L+) and T_{EM}/T_{RM} cells (CD44+CD62L-), communicated in the revised **Figures 4b-c** and **Supplementary Figure 7**. The lungs of the experienced mice did not possess CD62L⁺CD44⁺CD4⁺ T cells, confirming T_{CM} cells were not transiting through these extravascular non-lymphoid tissue compartments (**Fig. 4b**, **Supplementary Fig. 7a** and **page 13**), as was expected (Schenkel *et al*, *Immunity* 2014). The blood in the lungs of these mice contained circulating (i.v.CD45.2⁺) T_{CM} cells (clusters 9, 11, 16, and 20), and they did not differ between MHC-II^{fl/fl} and MHC-II^{ΔEpi} mice (**Figure 4c** and **Supplementary Figure 7b**), supporting our initial conclusion that LEC MHC-II has minimal effects on circulating memory T cells. To address whether LEC MHC-II constrains plasticity of pre-existing lineage-specified T_{RM} cells or if LEC MHC-II regulates development of plastic T_{RM} cells during their conversion from T_{eff} cells, we revised the manuscript to include data from additional new experiments. To better elucidate LEC MHC-II effects on plasticity of T_{RM} arising from T_{eff} cells, we generated CD4⁺ T_{RM} cells in C57BL/6J, MHC-II^{fl/fl}, and MHC-II^{ΔEpi} mice and examined if newly established CD4⁺ T_{RM} cells were exclusively lineage-specified (**Supplementary Figure 8a**). Consistent with the notion that these first CD4⁺ T_{RM} cells are not solely TH17-specified but are innately plastic, we observed heterogeneity and plasticity in CD4⁺ T_{RM} cells by day 56 (**Supplementary Figures 8b-c**), including a modest but significant expansion of T_H1/2/17 CD4⁺ T_{RM} cells in MHC-II^{ΔEpi} mice even without heterotypic memory recalls (**Supplementary Figures 8c-e**). These data suggested that LEC MHC-II might be regulating the development of multipotent CD4⁺ T_{RM} cells from tissue-infiltrating T_{eff} cells. We performed an additional set of experiments in order to generate CD4⁺ T_{RM} cells in MHC-II^{fl/fl} and MHC-II^{ΔEpi} mice before tamoxifen administration (and LEC MHC-II deletion), and then after MHC-II deletion from LECs we provided heterotypic memory recall exposures under constant pressure from the lymph node egress inhibitor FTY720 (**Supplementary Figure 9a**). This caused peripheral lymphopenia and prevented circulating CD4⁺ T cell access to the lungs (**Supplementary Figures 9b-d**). As such, only pre-established lung

CD4⁺ T_{RM} cells could contribute to the memory recall responses and plasticity depending on LEC MHC-II status. No differences in lung CD4⁺ T_{RM} cell phenotypes between MHC-II^{fl/fl} and MHC-II^{ΔEpi} lungs were observed (**Supplementary Figures 9e-h**). Altogether, these data support a model in which LEC MHC-II constrains plasticity during the development of CD4⁺ T_{RM} cells from T_{eff} cells in the primary immune response stages within the infected lungs.

b. The effect of class II loss on TRM plasticity is interesting. However, the mechanism (and whether for example the cells now interact with different lung class II+ APC) is not deciphered. A more mechanistic understanding would strengthen the paper.

We thank the reviewer for encouraging this line of investigation, which led to new experiments providing greater mechanistic insights into T_{RM} cell plasticity. Our initial studies had revealed that LEC MHC-II was essential for constraining T_{RM} cell plasticity and for surface PD-L1 expression on LECs. PD-L1 is a ligand for PD-1, and we observed high PD-1 expression on T_{RM} cells (**Figure 3d** in the revision), which is characteristic of CD4⁺ T_{RM} cells but has never been functionally defined on these cells. Putting these observations together suggested the novel hypothesis that LEC PD-L1 might engage PD-1 on T cells to constrain their plasticity, a cell-cell interaction that would be severed by LEC MHC-II loss. To test whether the PD-L1 ligand PD-1 constrains the plasticity of lung T_{RM} cells, we generated CD4⁺ T_{RM} cells in PD-1 knockout (PD-1^{-/-}) mice by instilling Sp19F followed by recurrent memory recalls with heterotypic infections and in-depth phenotyping of CD4⁺ T_{RM} cell plasticity at day 85 (**Figure 6e**). Consistent with our hypothesized mechanism, *Spn*-experienced PD-1^{-/-} lungs exhibited significantly greater CD4⁺ T_{RM} cell plasticity (as evidenced by more triple-positive T_H1/2/17 T_{RM} cells) and an enhanced multipotency index, similar to the T_{RM} cells from MHC-II^{ΔEpi} lungs which lacked PD-L1 on their LECs. These new experiments were communicated in the new **Figures 6e-g, Supplementary Figure 18** and **pages 20, 21, 23, 26, and 49** in the revision, and they support a novel and newly proposed mechanism for regulation of T_{RM} cell plasticity in the lung. We now propose that LEC MHC-II facilitates PD-L1 expression on LECs so that LEC PD-L1 can trigger PD-1 on T cell surfaces which then harnesses the phenotypic options of resulting T_{RM} cells.

9. In Figures 5 and 6, the effects of MHCII deficiency in LEC on TRM cells have been evaluated by cytokine production and innate immune instruction. However, a key feature of Sp infection is also iBALT formation, which can suggest the pathogen-specificity of the CD4+ TRM cells. In evaluating the effects of LEC MHCII on CD4+ TRM maintenance and recall function, pathogen-specificity is an important factor. An experiment to address this specificity question will strengthen the study.

We appreciate 2 concerns in this comment, the connections of our findings to iBALT and to pathogen-specific lung T_{RM} cells. iBALT is relevant to some types of

pulmonary challenge, but not the *Spn* infections in the manuscript under consideration. iBALT forms in mouse lungs after severe challenges such as from the PR8 strain of influenza A virus, but the Sp19F infections used here to elicit lung-resident memory do not cause iBALT formation, based on the absence of large lymphoid aggregates in the lung, the absence of a follicular organization to the T and B cells that do deposit in these lungs, and the absence of PNAd⁺ high endothelial venules with or near the T_{RM} and B_{RM} cells in these lungs (Smith *et al*, *Mucosal Immunology* 2018; Barker *et al*, *J Clin Invest* 2021). Consistent with this, the present studies (**Figure 3e** of the revision) do not suggest that lung CD4⁺ T cells were organized into iBALT-like structures. Furthermore, iBALT associates with dysplastic repair and aberrant subsets of LECs that are not found in healthy lungs, including keratin 5⁺ epithelioid cells in alveolar regions (as elucidated in multiple studies by Hal Chapman, Andy Vaughan, Barry Stripp, and others) plus perpetually inflamed and activated epithelial cells in the airways (elucidated by studies from Nick Heaton). The *Spn* infections used here do not cause such epithelial dysplasia, allowing us to focus on normal LECs and their responses to acute and resolved infections. It is possible that MHC-II is expressed and antigen presentation is accomplished by the aberrant epithelial cells of iBALT-containing lungs after PR8 infection, which would be an interesting focus for another paper but is beyond the scope of the present manuscript. Regarding *Spn* specificity of the lung T_{RM} cells, we addressed this using new experiments and *ex vivo* cultures to examine antigen specificity of T_{RM} responses. We found that lung T_{RM} cells from MHC-II^{ΔEpi} mice responded differently due to genotype (with less IFN_γ induction in the MHC-II^{ΔEpi} mice) when killed *Spn* was provided as antigen, whereas there were no genotype-specific effects on T_{RM} cells when an irrelevant antigen (ovalbumin) was used instead (**Supplementary Figure 12** and **pages 17** in the revised version). It was also readily apparent that lung T_{RM} cells made orders of magnitude more IFN_γ and IL-17A when killed *Spn* was used as the antigen instead of the irrelevant ovalbumin antigen (**Supplementary Figure 12** in the revised version). These new experiments strengthen the study by demonstrating LEC MHC-II-dependent effects on antigen-specific lung T_{RM} cells.

10. Figure 6C: The changes in distribution of cells are shown. Do absolute numbers of the different cell types change to a similar degree?

We revised as requested to communicate absolute numbers of the different cell types (**Supplementary Figures 14 and 15** in the revision). The majority of the cell types changed to a similar degree and in a similar direction when analyzed using numbers or fractions. We hold higher confidence in the data expressed as fractions since mice of both sexes were included in our study, necessitating that individual animals varied considerably across body and lung sizes across the 6 month long experimental regimen, confounding interpretations related to absolute numbers. Absolute numbers provide helpful perspective on amplitude of the immune responses, but fractions of immune cells better represent the phenotypes of these immune responses. The revised manuscript includes and fully presents both sets of data, analyzed as fractions (**Figure 5c** and

Supplementary Figure 15a in the revision) and as absolute numbers (**Supplementary Figures 14 and 15b** in the revision). Making the data available in both formats will allow readers to make their own most informed conclusions.

11. Based on Figure 7, the authors showed a strong correlation between MHCII and PD-L1 expression in human lungs and inhibition of surface presentation of PD-L1 in LEC lacking MHCII in the murine model. However, direct evidence linking the absence of MHCII to the expression and localization of PD-L1 is lacking, leaving it unclear whether the differential expression and localization of PD-L1 is a direct or indirect effect of MHCII expression in LECs. A more elaborate experiment exploring the molecular factors would add significantly to this observation.

We wholeheartedly agree and are actively pursuing this direction to investigate molecular factors mediating the dependence of PD-L1 on MHC-II in LECs. We are planning to dedicate an entire separate paper to this question. We are in early stages of defining which specific factors are involved. For the current manuscript, we revised to expand upon these PD-L1:MHC-II connections in LECs by showing that the relationship applies across diverse stimuli, and that alveolar epithelial cell surface PD-L1 is strictly dependent on endogenous MHC-II even when stimulated by the non-infectious immune mediator IFN- γ (**Supplementary Figure 17** and its legend as well as **pages 19, 27, 32** in the revision).

Minor Comments

1. The MHC^{high} LEC exhibit some features of lung progenitor cells. More comprehensive characterization of the “stem-ness” of these cells will be of interest.

Here, can the authors comment on how they think these cells compare to the AT2-signalling stem cells reported in Travaglini et al, Nature, 2020.

We share the reviewer’s interest in the possibilities that our SPC^{low}MHC^{high} LECs may have characteristics reminiscent of lung stem cells. Progenitor and stem cell biology in the lungs is a highly dynamic and rapidly progressing area of study, with distinct stem and progenitor cell subsets being defined and refined frequently and heterogeneously across groups. Travaglini *et al* discovered a subset of AT2 cells that could function as stem cells in human lungs, present in ten-fold lower abundance than traditional AT2 cells. Our SPC^{low}MHC^{high} LECs were comparable in number to traditional AT2 cells in murine lungs, differing in relative abundance from the cells of Travaglini *et al*, but such relative cell numbers are collection methods-dependent and difficult to compare across studies due to apparent as well as unappreciated differences in lung digest protocols. Among the markers we measured in SPC^{low}MHC^{high} LECs, the human AT2-signalling cells of Travaglini *et al* compared to canonical AT2 cells express lower levels of SPC (SFTPC gene), lower levels of CDKN1A, and higher levels of

AQP5. The SPC finding corresponds across our cells with Travaglini's AT2-s cells, but the other two markers trend in opposite directions. We suspect therefore that our SPC^{low}MHC^{high} LECs in mouse lungs may be distinct from the AT2-signalling cells of human lungs defined by Travaglini. We revised the manuscript to reference the Travaglini paper and suggest that a comparison of cells will be merited (**page 21** in the revision). Also related to "stem-ness" of the novel cells we described here, we considered the possibility that our SPC^{low}MHC^{high} LECs might overlap with bronchioalveolar stem cells (BASCs) in mouse lungs, both of which contain transcripts for *Sftpc* but at lower levels than in AT2 cells and for *Scgb1a1* but at lower levels than in club cells. We revised our manuscript to include a new collaboration involving BASC-tracer transgenic mice. These new experiments demonstrated that our SPC^{low}MHC^{high} LECs were distinct from BASCs (**Figures 1g-h** and text on **pages 5-6, 26, 28, and 45** in the revision). It remains to be determined whether or not our SPC^{low}MHC^{high} LECs have stem cell characteristics, but we to date have not been able to match them faithfully to other stem cells that have been characterized within the lungs.

2. In Figure S1B: the gating strategy for DC would seem to identify a subset of DCs (there are CD24neg cDC2 in the lung) and potentially include B cells. While this may not be critical for a comparison of class II levels with LEC, have the authors used other markers to determine the identity of the gated cells?

We thank the reviewer for pointing out that our strategy did not include all or only dendritic cells. Unfortunately, there were not more detailed markers in those studies for more precisely determining identities of those gated cells. Recognizing that the gate omitted CD24⁻ cDC2s and could have included B cells (though rare in naïve mouse lungs; Barker *et al*, *J Clin Invest* 2021), we revised the text to more explicitly state that CD45⁺CD24⁺MHCII⁺ APCs were the positive control comparators in this experiment (**page 5** in the revision and **Supplementary Figure 1b**).

3. Figure 2F-K: statistical annotation is missing from plots. Also, please convert the y-axes of Figure 2F and 2I to decimal to be consistent with the other data in the same panel.

Revised as suggested. For 2F-K, there were so many statistical comparisons of possible interest that the revision now provides a separate set of tables for communicating the results of these many analyses (**Supplementary Table 1-6** in the revision). The y-axes were converted as requested (**Figures 2g** and **2** in the revision).

4. Figure S4: Legend for panel C is missing, so C-E letters in legend are also incorrect.

We thank the reviewer for catching these errors. **Supplementary Figures 2d, 2e, 5c, 5d** and their legends were revised to correct these errors.

5. Figure S5B: is the cell/label correct as MHChi Club?

We thank the reviewer for catching this error. **Supplementary Figure 6f** was revised to correct.

6. Fig 3D: the results for T_{eff} are missing. This effects the statement that only TRM are still PD-1+ at day 35.

Our interpretation is that no T_{eff} cells are present at the day 35 time-point. Both T_{eff} cells and T_{RM} cells are CD62L-CD44+CD11a+CD69+. Based on the set of markers available to the field (and included in our study), we cannot decisively discriminate between T_{eff} cells and T_{RM} cells, as communicated in **Figure 3a**. Nevertheless, given that *Spn* experienced lungs have returned to homeostasis on Day 35 after Sp19F infection and T_{eff} cells are actively proliferating and producing cytokines in response to active stimulation, we chose to identify CD62L-CD44+CD11a+CD69+ T cells on Day 35 as T_{RM} cells. These cells all expressed high levels of PD-1 and CD69, and T_{RM} cells are the only CD62L-CD44+ T cell subset expressing high levels of PD-1 and CD69 in resting tissues (e.g., as reviewed in Szabo *et al*, *Science Immunology* 2019).

7. Figure 3i: Is the IF image from day 56, but the quantitation from day 35? Legend also says one-way ANOVA, but the comparisons are each between only 2 groups.

We thank the reviewer for catching this error. **Figure 3h** in the revision was revised to correct the inconsistent labeling. The statistical comparison performed was one-way ANOVA to account for the multiple comparisons, but we had neglected to identify the *post hoc* analyses that had been applied in order to make the specific comparisons of interest and communicated in the figure. This was corrected in the revision (**pages 46-47**).

8. Line 123 Should be, which are...

Revised as suggested.

9. Line 181 The modulation of APC molecules on LECs on day 8 was microbe-induced,

Revised as suggested.

10. Line 196 snapshots of the dynamics

Revised as suggested.

11. Line 705 needs editing

Revised as suggested.

Reviewer #3 (Remarks to the Author):

The manuscript “Epithelial MHC-II governs CD4+ TRM cells and barrier immunity” examines the impact of MHC-II expression on lung CD4+ TRM location, plasticity and the lung immune landscape. Using SPC-GFP mice, electron microscopy, and MFSC analysis, the authors characterized lung epithelial cells, including a population of MHC-II^{high} LEC with a novel phenotype distinct from both club cells and traditional AT2 cells. They demonstrated that all LEC exhibit antigen presenting ability, but that following Spn infection, costimulatory and coinhibitory molecule expression by LEC is dynamic and cell-type specific. The authors suggest that lung epithelial MHC-II facilitates CD4+ TRM formation, since MHC-II Δ epi mice have smaller CD4+ TRM niches along the airway epithelium and within bronchovascular interstitium. Additionally, LEC-MHCII also regulates the phenotype of CD4+ TRM. Again, using MSFC analysis, the authors found that following repeated exposure to a serotype-mismatched strain of Spn, CD4+ TRM are heterogeneous and express multiple lineage defining transcription factors. Additionally, CD4+ TRM expressing multiple LDTFs increase in the absence of LEC MHCII. In contrast, circulating memory CD4+ T cells do not exhibit as much plasticity. Following re-stimulation, T cell clusters express fewer LDTFs, and in the absence of LEC MHC-II, lung CD4+ T cells have increased Th17 response, but decreased Th1 response correlating with dysregulation in immune barrier landscape. Finally, the authors noted similarities in the immune dysregulation of infected MHC-II Δ epi mice and checkpoint blockade immunotherapy patients. They demonstrated that LEC MHC-II expression correlates with PD-L1 expression across all LEC cell types in mice as well as by LEC in humans. Additionally, LEC from MHC-II Δ epi lack PD-L1 expression, suggesting a lockstep between MHC-II and PD-L1 on LECs. In the absence of MHC-II, PD-L1 was trapped in the intracellular compartment and degraded. The authors present novel data examining the effect of lung epithelial cell MHC-II expression on CD4+ TRM localization, plasticity and immune response following challenge that may be of importance to the general community. The is a well-executed study. However, a few comments and suggestions might help to support the conclusions and clarify the significance of these findings:

1. The authors use H2-Ab1/fl mice crossed to Nkx2.1Cre-ERT2 mice to examine the requirement of MHCII expression by LEC. The authors demonstrate that MHC-II remains intact on CD45+ leukocytes following administration of tamoxifen. Is MHCII expressed by other CD45- cells at baseline or following activation in these mice following administration of tamoxifen?

We thank the reviewer for this important question. We performed new experiments to address it. As suggested, a sub-population of CD45(-) cells in the lungs does express MHC-II, but this non-epithelial MHC-II expression is unaffected by tamoxifen-driven Nkx2.1-CreERT2 targeting of the H2-Ab1 locus. The lung CD45(+) cells were also unaffected by the gene targeting. These new data help demonstrate that the gene targeting approach used herein does not

have off-target effects on MHC-II in non-epithelial cells of the lung. The manuscript was revised to include these new data in **Supplemental Figure 2e** and its legend and **page 7** of the revision.

2. The section title “a novel population of LECs” might be a bit strong. Might these cells be a subset?

Agreed. We revised the section title to omit claims of a novel sub-population and instead identified cells of interest by their measured SPC and MHC characteristics.

3. Does the plasticity of CD4⁺ TRM depend on the Spc serotype used for infection? For example, in Figure 4, if memory recalls were performed with Sp35B rather than with Sp23A, but analyses were still performed at day 105, would the CD4⁺ TRM still exhibit the same plasticity? Also, what is LDTF expression by TRM on day 56 before memory recalls?

It seems highly unlikely that CD4⁺ T_{RM} cell phenotypes might depend on *Spn* serotype. *Spn* serotype is determined by capsular polysaccharides that are recognized by serotype-specific antibodies. Since peptides and not sugars are antigens for CD4⁺ T cells, different capsular sugars would not be expected to directly influence CD4⁺ T cells, unless variations in the enzymes generating those sugars change peptides that are pivotal TCR ligands. Comparing one or more isolates of Sp23A with one or more isolates of Sp35B would be insufficient to make conclusions about serotype, as different isolates of *Spn* (whether they are of matched or differing serotypes) have many and diverse genomic differences which sort independently from serotype. *Spn* strains have been developed for using Janus cassette systems to facilitate capsule switching within an otherwise fixed genome, which would be necessary to test effects of serotype specifically. However, with ≥100 serotypes of pneumococcus so far identified, it would require an inordinate amount of work to test whether some or any capsule(s) or their polysaccharide-modifying enzymes could possibly be capable of altering CD4⁺ T cell responses in the multi-months long T_{RM} plasticity assays. To our knowledge, no precedent would suggest it, limiting the rationale for pursuing this line of investigation. Mismatching serotypes was key to the studies here communicated in order to avoid the generation of serotype-specific antibodies, which are so highly opsonophagocytic and effective for host defense that they have confounding effects like obviating the need for heterotypic T cells. However, the specific serotypes should be irrelevant, as long as they are serially mismatched, which has been confirmed so far in all of our studies which have used a variety of different serotypes with overlapping effects relating to heterotypic T cell immunity (Smith *et al*, *Mucosal Immunology* 2018; Shenoy *et al*, *Mucosal Immunology* 2020; Barker *et al*, *J Clin Invest* 2021). The question about timing and the possibility of earlier development of LDTF-related plasticity (at day 56) is provocative and led to a series of additional experiments. We comprehensively phenotyped CD4⁺ T_{RM} cells in *Spn*-experienced C57BL/6J,

MHC-II^{fl/fl}, and MHC-II^{ΔEpi} mice on Day 56 (**Supplementary Fig. 8a** in the revision). Consistent with our data in the more extensively experienced mice, we found that CD4⁺ T_{RM} cells were diverse and innately plastic (**Supplementary Figure 8b-c** in the revision). There was evidence of modest but significant expansion of LDTF plasticity (T_H1/2/17 skewed T_{RM} cells in gray in **Supplementary Figure 8d**) due to LEC deletion of MHC-II, even before memory recalls. Altogether, results from the many studies included in the manuscript support the idea that LEC MHC-II is critical for constraining the plasticity of recent emigrants which get activated in the lung to differentiate towards the T_{RM} phenotypes.

4. Can memory recall induce formation of non-antigen-specific lung CD4+ TRM? In Figure 5H, if cells are stimulated with antigen-pulsed APCs rather than PMA and ionomycin, is the same pattern of cytokine secretion observed?

To test if lung CD4⁺ T_{RM} cells at day 105 are *Spn*-specific and to determine if their cytokine secretion pattern was similar to PMA/ionomycin treatment, we stimulated single cell suspensions of day 105 MHCII^{fl/fl} and MHCII^{ΔEpi} lungs with beta-propiolactone killed Sp35B (**Supplementary Fig. 12** in the revised version). We also included parallel cultures where these cells were stimulated instead with ovalbumin as an irrelevant antigen. Consistent with the T_{RM} cells being *Spn* specific, we saw strong production of both IFN-γ and IL-17A after *Spn*-stimulation, suggesting polyfunctionality. These lung cell responses to *Spn* were orders of magnitude greater than the responses from blood cells, and orders of magnitude greater than the responses of the same lung cells to ovalbumin. After the *Spn* stimulation, but not the ovalbumin stimulation, the T_{RM} cells from lungs without LEC MHC-II were skewed away from IFNγ, again similar to the observations with PMA/ionomycin. These results on the whole lead us to infer that the repeated memory recalls as here delivered led to lung T_{RM} that were primarily antigen-specific, and the T_{RM} cell biology revealed from PMA/ionomycin stimulations was relevant to antigen-specific responses from the T_{RM} cells.

Minor comments:

1. In Figure 1B, it might be nice to have a legend for the flow plot indicating that the dark blue population is dendritic cells.

For **Figure 1b**, the flow plot is matched by colors to the associated graphed MFI data, with the dark blue now revised to indicate the blue data as coming from APCs in both the flow plot and the bar graph. The text on **page 5** was also revised to specify that these APCs were CD45⁺CD24⁺MHC-II⁺ cells.

2. In Figure 3A, was FoxP3 analyzed? Could the CD25+ cells be Treg?

Unfortunately, FoxP3 was not included as part of the flow panel used to generate Figure 3a. Given that both CD25 and CD69 signal activated cells during an active

immune response such as on days 7-14 in the infection protocol (**Figure 3a** in the revision), the absence of Foxp3 leaves the possibility of T_{reg} cells difficult to gauge. At day 35, after the lung is back to rest and adaptive immune responses have contracted, CD69 then marks T_{RM} cells and very few express CD25. Furthermore, in prior studies, these day 35 cells do not respond to stimulation with PMA/ionomycin or antigen presentation from killed *Spn* with IL-10 elaboration (Smith et al, *Mucosal Immunology* 2018). Because we cannot discriminate T_{reg} from T_{eff} or T_{RM}, we revised the manuscript in **Figure 3a** to refer to CD25+ cells as being CD25+ T_{eff} phenotype at earlier time-points and CD25+ T_{RM} phenotype at later timepoints.

3. Is it possible that contraction of CD4+ T cells in the parenchyma results from lack of costimulatory receptors by LEC?

Yes, this certainly seems like a reasonable possibility to us. While our manuscript has generated new information about mechanisms that establish and program the CD4+ T_{RM} cells in the bronchovascular niches of the lung where they are found, we very much agree that the mechanisms responsible for preventing the seeding of CD4+ T_{RM} cells in the alveolar septae remain unknown and an important area for future research.

4. It might be helpful for readers without a lung biology background to have a supplementary figure with a schematic mapping the locations of LEC cell types (perhaps with an indication of their MHC and costimulatory expression) and CD4+ T cells over time following infection.

This was a terrific idea, and we hope the reviewers and readers will enjoy this new figure as much as we enjoyed making it. In response to this suggestion, we revised the manuscript to include a schema that maps the locations of LEC cell types with an indication of their MHC-II and costimulatory expression (**Supplementary Figure 19a** in the revision), highlights CD4+ T cell dynamics over time following infection (**Supplementary Figure 19b** in the revision), and indicates constraint of CD4+ T cell plasticity as a major role of LEC MHC-II (**Supplementary Figure 19c** in the revision). We hope this revision will help clarify and emphasize major findings from this manuscript related to how epithelial cell biology and adaptive immunity are functionally intertwined in the lungs.

REVIEWERS' COMMENTS

Reviewer #1 (Remarks to the Author):

The revised manuscript is considerably improved and addresses all my concerns. It would have been good to have marked the changes in the text!

Reviewer #2 (Remarks to the Author):

The authors have done an outstanding job responding to the concerns of the reviewers. I believe the revised version of the manuscript is acceptable for publication, with a few minor corrections as outlined in response to the authors below.

Reviewer #3 (Remarks to the Author):

All comments have been addressed.
The only small suggestion that the authors might consider would be to use the same y-axis scale for all of the panels in Figure 2c.

Reviewer #1 (Remarks to the Author):

The revised manuscript is considerably improved and addresses all my concerns. It would have been good to have marked the changes in the text!

We thank the reviewer for their positive comments and have included a track edited version of the manuscript with the necessary editorial edits made.

Reviewer #2 (Remarks to the Author):

The authors have done an outstanding job responding to the concerns of the reviewers. I believe the revised version of the manuscript is acceptable for publication, with a few minor corrections as outlined in response to the authors below.

We thank the reviewer for their positive comments. We could not find the list of minor corrections that Reviewer #2 raised and assume that they were included in the list of final revisions requested by the editor. The track-changes version of the manuscript shows revisions to all suggestions forwarded by the editor.

Reviewer #3 (Remarks to the Author):

All comments have been addressed. The only small suggestion that the authors might consider would be to use the same y-axis scale for all of the panels in Figure 2c.

We thank the reviewer for their positive comments. Recognizing that uniform y-axis scales facilitates comparing the different cell-types to each other, we have generated and here include an 'Alternate Figure 2' with y-axis edited as per Reviewer #3's request. We strongly prefer the original "Figure 2" for publication since it **highlights the more important comparisons**, between cre+ vs cre- cells of the same type within each panel (rather than between one cell-type and another across panels). The original Figure 2c makes it much more evident that all lung epithelial cells (LECs), whether airway or alveolar, are capable of antigen presentation (demonstrated by the reduced CD4+ T cell activation in cre+ MHC-II^{ΔEpi} mice compares to their cre- controls). In light of the reviewer's concern, we added a sentence to the revised Figure Legend in order to emphasize the different scales within the figure and to clarify that the different cell-types had differing abilities to function as APCs. While we feel that the original Figure 2c is far superior to the version suggested by the reviewer, we include both here so that the editor can compare and decide which is preferable. We will accept and respect the editor's decision.